# Fast attention mechanisms: a tale of parallelism

**Jingwen Liu**
Columbia University
New York, NY
jingwenliu@cs.columbia.edu

**Hantao Yu**
Columbia University
New York, NY
hantao.yu@columbia.edu

**Clayton Sanford**
Google Research
San Francisco, CA
chsanford@google.com

**Alexandr Andoni**
Columbia University
New York, NY
andoni@cs.columbia.edu

**Daniel Hsu**
Columbia University
New York, NY
djhsu@cs.columbia.edu

## Abstract

Transformers have the representational capacity to simulate Massively Parallel Computation (MPC) algorithms, but they suffer from quadratic time complexity, which severely limits their scalability. We introduce an efficient attention mechanism called Approximate Nearest Neighbor Attention (ANNA) with sub-quadratic time complexity. We prove that ANNA-transformers (1) retain the expressive power previously established for standard attention in terms of matching the capabilities of MPC algorithms, and (2) can solve key reasoning tasks such as Match2 and $k$-hop with near-optimal depth. Using the MPC framework, we further prove that constant-depth ANNA-transformers can simulate constant-depth low-rank transformers, thereby providing a unified way to reason about a broad class of efficient attention approximations.

## 1 Introduction

The transformer [57] has become the dominant neural architecture in deep learning due to its ability to select and compose complex information structures from large inputs [1, 19, 24], which in turn enables capabilities such as "in-context learning" [13] that are crucial for downstream applications. At the core of transformers is the attention mechanism, which leverages parallelism to gain important advantages over non-parallel architectures (e.g., recurrent nets), including training stability [46] and representational power [11, 31, 41, 45, 52, 55, 60]. One recently highlighted advantage comes from a coarse theoretical relationship between transformers and the Massively Parallel Computation (MPC) model [33] that captures the power of large-scale distributed computing frameworks like MapReduce [17]: efficient MPC algorithms can be simulated by small-size transformers, and vice versa [53, 55]. This correspondence suggests that a broad class of computational tasks can be efficiently solved by transformers.

Despite these advantages, transformers suffer from a key limitation: the quadratic time complexity of (standard) attention with respect to input size, which is likely to be unavoidable in the worst-case [2, 3, 34]. To address this limitation, a growing body of work proposes alternatives to the attention mechanism that are more computationally efficient (e.g., sub-quadratic time). These alternatives employ a variety of techniques, ranging from low-rank approximation [8, 14, 16, 32, 58] to efficient nearest neighbor search [27, 35, 51, 64]. Although many of these techniques show promising empirical performance, it is unclear whether they preserve the representational advantages of attention.

39th Conference on Neural Information Processing Systems (NeurIPS 2025).

In this work, we study a particular efficient (sub-quadratic time) attention mechanism, called Approximate Nearest Neighbor Attention (ANNA), and we prove that ANNA retains key representational advantages of standard attention over non-parallel architectures. We additionally prove that ANNA-transformers can simulate a class of attention mechanisms based on low-rank approximation. In doing so, we provide a unified way to reason about low-rank and nearest neighbor approaches to efficient attention.

## 1.1 Standard transformers and MPC

Prior work [53, 55] established a coarse relationship between standard transformers and MPC by proving the following (for any constant $\delta \in (0, 1)$):

1. For any MPC algorithm with inputs of size $N$ that uses $R$ rounds (of computation and communication), $O(N)$ machines, and $O(N^\varepsilon)$ words of local memory per machine (for some constant $\varepsilon \in (0, 1)$), there is an equivalent transformer of width $O(N^{\varepsilon+\delta})$ and depth $O(R)$. (By width, we mean number of heads per layer times the embedding dimension.)

2. For any transformer operating on inputs of size $N$ that has $L$ layers and width $O(N^\varepsilon)$ (for some constant $\varepsilon \in (0, 1)$), there is an equivalent MPC algorithm that uses $O(L)$ rounds, $O(N^2)$ machines, and $O(N^{\varepsilon+\delta})$ words of local memory per machine.

Notice that the "loss" in the number of rounds or depth is only a constant factor, and the "loss" in the local memory size or width is only a $O(N^\delta)$ factor (where $\delta > 0$ can be arbitrarily small). Consequently, these results were sufficient to give new results about transformer representational power (e.g., for various graph reasoning tasks [53, 55]), and also give a (conditional) logarithmic-depth lower bound for transformers that solve a multi-hop reasoning problem called $k$-hop [55].

However, there is a important gap in the MPC algorithm for simulating a transformer (Item 2 above): *it may use up to $N^2$ machines*. This gap suggests the possibility that transformers are strictly more powerful than MPC algorithms, so the relationship established in prior work is very coarse. Moreover, this gap is likely to be inevitable. Indeed, when the local computation in an MPC algorithm is fast (say, polynomial-time), and the local memory size is $N^\varepsilon$ for small $\varepsilon \in (0, 1)$, then a single round of MPC on $M$ machines can be simulated on a sequential machine in roughly $M N^{O(\varepsilon)}$ time. However, evaluation of attention over $N$ inputs is believed to require at least $N^{2-\delta}$ time (for every constant $\delta > 0$) on a sequential machine [2, 3, 34]. So any such MPC algorithm that simulates attention in $\tilde{O}(1)$ rounds should use $N^{2-\delta-O(\varepsilon)}$ machines.

This gap naturally leads us to the following question: Is there an alternative attention mechanism that more tightly captures the computational power of MPC?

## 1.2 Our contributions

We prove that ANNA-transformers more sharply capture the power of MPC algorithms than standard transformers do, and in particular avoid the aforementioned gap in prior works' characterization of transformers using MPC [53, 55]. The core ANNA mechanism is designed to perform $c$-approximate nearest neighbor search [30, 38] for a given set of $N$ "queries" against a database of $N$ "key-value" pairs. Here $c > 1$ is a parameter of the ANNA mechanism that, for the purpose of this present informal description, should be thought of as being at least a large positive constant (e.g., $c \geq 10$). Throughout, $N$ denotes the input size.

**Theorem 1.1** (Informal version of Theorems 4.1 and 4.2). *Fix any constants $\varepsilon, \delta \in (0, 1)$ and ANNA parameter $c > 1$. For any $R$-round MPC algorithm with $N^\varepsilon$ words of local memory, there is an equivalent $O(R)$-layer ANNA-transformer of width $O(N^{\varepsilon+\delta})$. For any $L$-layer ANNA-transformer with width $N^\varepsilon$, there is an equivalent $O(L)$-round MPC algorithm that uses $N^{\varepsilon+\delta}$ words of local memory and $N^{1-\delta+O(1/c^2)}$ machines.*

Observe that Theorem 1.1 essentially mirrors Items 1 and 2 in Section 1.1 except for the resources needed by the MPC algorithm in Item 2: the local memory stays the same, but the number of machines required can be strongly sub-quadratic (and in fact, close to linear) in $N$. This gives a positive answer to the question at the end of Section 1.1.

We also study efficient attention mechanisms based on low-rank approximation, and by leveraging Theorem 1.1, we prove that ANNA-transformers have at least the same representational power.

**Theorem 1.2** (Informal version of Theorem 4.4). *For any L-layer low-rank attention transformer, there is an equivalent $O(L)$-layer ANNA-transformer (of comparable width).*

Finally, we illustrate the power of ANNA-transformers on two concrete reasoning tasks (Match2 [52] and $k$-hop induction heads [12, 55]). We give theoretical constructions of ANNA transformers for these tasks that nearly match the efficiency achievable by standard transformers (Theorems 5.2 and 5.6), and we also show empirically that ANNA-transformers can be trained to approximately solve these tasks (Section 5.3).

### 1.3 Other related works

The representational power of standard attention is relatively well understood from a variety of perspectives. This includes: universal approximation properties in the large depth/width limit [22, 44, 59, 63], ability to recognize formal languages [10, 25, 41, 56, 62], computational bounds in terms of circuit classes [28, 39, 41, 43] and other parallel computation models (discussed below) [53, 55], and bounds for specific compositional tasks [12, 15, 36, 41, 49, 54]. In general, these and other prior works do not consider the representational power of transformers based on efficient (sub-quadratic time) alternatives to attention. The exceptions are works that give lower bounds for low-rank attention and sparse attention [55, 61], sequential architectures [11, 31, 45, 52, 60], more generally, any mechanism that can be evaluated in sub-quadratic time [2, 3, 34]. What is missing in these prior works, however, is a characterization (and, in particular, upper bounds) for an efficient architecture.

Massively Parallel Computation (MPC) [5, 9, 20, 23, 29, 33] is a model of parallel computing intended to capture the power of MapReduce [17] and other large-scale distributed computing frameworks. Many works, including those that originally helped to define the MPC model, gave efficient algorithms for a variety of basic data and graph processing tasks. A connection between MPC and circuit models was given by [50] and used to formalize a barrier on proving lower bounds for certain graph problems (e.g., connectivity) in MPC. Nevertheless, certain conjectured lower bounds in MPC are widely believed (e.g., the 1-vs-2 cycle conjecture [29]) and have been used to establish conditional lower bounds for other problems [21]. The same conjectured lower bounds have also been used to establish depth lower bounds for transformers via the aforementioned relationship between MPC and transformers [53, 55].

Our main result for ANNA-transformers has an analogue in the context of message-passing graph neural networks (GNNs). The computational power of such GNNs is characterized by the CONGEST model of distributed computing [48] (a refinement of LOCAL [7, 40, 47]). This fact was established by [42] and in turn used to give upper- and lower-bounds on the size of GNNs needed to solve various graph problems (e.g., subgraph detection). The GNN/CONGEST equivalence is almost immediate, since the communication network is fixed by the input graph in both models, and the model definitions are semantically and syntactically similar. In contrast, while the layered representations of transformers are reminiscent of the alternation between communication and computation rounds in MPC, the "communication patterns" themselves are dynamic, and this dynamism greatly complicates the simulation of MPC algorithms by either standard transformers or ANNA-transformers.

The ANNA mechanism that we study is inspired by many prior efficient attention mechanisms [e.g., 27, 35, 64] based on locality-sensitive hashing [30], which is one of the key techniques for nearest neighbor search. Some of these mechanisms have guarantees about the quality of approximation under structural assumptions on the attention matrix they are meant to approximate. The motivation of our analysis is largely orthogonal: we instead seek to characterize the representational power of ANNA-transformers in terms of other well-understood models of parallel computation. We further compare the capabilities of LSH-based efficient attention to other sub-quadratic alternatives, including those based on low-rank approximations of self-attention matrices [16, 32, 58].

## 2 Preliminaries

### 2.1 Standard attention and transformers

We first define the (standard) attention mechanism and transformers.

**Definition 2.1** (Attention). *A* (standard) attention head $\text{Attn}_{Q,K,V}$ *is specified by query, key, value embedding functions* $Q, K, V : \mathbb{R}^d \to \mathbb{R}^m$. *On input* $X \in \mathbb{R}^{N \times d}$, *it computes*

$$\text{Attn}_{Q,K,V}(X) := \text{softmax}(Q(X)K(X)^\top)V(X) \in \mathbb{R}^{N \times m},$$

*where* $Q$, $K$, $V$, *and* softmax *are applied row-wise. We say* $N$ *is the* context length, *and* $m$ *is the* embedding dimension. *The rows of* $Q(X)$ *(resp.,* $K(X)$, $V(X)$*) are the* queries *(resp.,* keys, values*). If* $q_i = Q(X)_i$, $k_j = K(X)_j$, *and* $v_j = V(X)_j$, *then the* $i$*-th row of* $\text{Attn}_{Q,K,V}(X)$ *is*

$$\text{Attn}_{Q,K,V}(X)_i = \sum_{j=1}^{N} w_{i,j} v_j, \quad \text{where} \quad w_{i,j} = \frac{\exp(\langle q_i, k_j \rangle)}{\sum_{j'=1}^{N} \exp(\langle q_i, k_{j'} \rangle)}.$$

*An* $H$*-headed* attention layer $f : \mathbb{R}^{N \times d} \to \mathbb{R}^{N \times d}$ *consists of attention heads* $(\text{Attn}_{Q_h,K_h,V_h})_{h=1}^{H}$ *and* $m \times d$ *matrices* $(W_h)_{h=1}^{H}$; *it computes* $f(X) := \sum_{h=1}^{H} \text{Attn}_{Q_h,K_h,V_h}(X)W_h$.

**Definition 2.2** (Transformer). *An* $L$*-layer* transformer $\mathcal{T}$ *is specified by attention layers* $f_1, \ldots, f_L : \mathbb{R}^{N \times d} \to \mathbb{R}^{N \times d}$ *and an output function* $\psi : \mathbb{R}^d \to \mathbb{R}^d$. *Given input* $X \in \mathbb{R}^{N \times d}$, *define*

$$X^{(0)} := X \quad \text{and} \quad X^{(\ell)} := f_\ell(X^{(\ell-1)}) \quad \text{for } \ell = 1, \ldots, L.$$

*The output of* $\mathcal{T}$ *on input* $X$ *is* $\psi(X^{(L)})$ *with* $\psi$ *being applied row-wise.*

In this paper, we consider the context length $N$ (the number of input tokens) as the principal scaling parameter. This reflects the modern paradigm of long-context LLMs, where the context-length can exceed $10^6$ [19], enabling book-length textual inputs. Consequently, we typically want the size parameters $m$, $H$, and $L$ to be sub-linear in $N$ (and $L$ ideally constant). The $O(N^2)$ runtime of attention therefore remains the main bottleneck of the transformer architecture.

Following [55], we allow the element-wise operations $(Q_h, K_h, V_h, \psi)$ to be arbitrary functions [1] (limited only by bit-precision; see [55, Appendix A.1]). Therefore, we can view a transformer as a computational model that alternates between arbitrary per-token computation, and communication between tokens. This motivates a connection to massively parallel computation, defined next.

## 2.2 Massively Parallel Computation

The Massively Parallel Computation (MPC) framework [29] models computation on large inputs by a distributed computing system that alternates between rounds of local computation and rounds of restricted all-to-all communication. We formally state the definition of MPC protocols (with sublinear memory) as follows.

**Definition 2.3** (MPC). *For constants* $\gamma, \varepsilon > 0$, *an* $R$*-round* $(\gamma, \varepsilon)$*-MPC protocol specifies the following computation on inputs of* $N$ *words (where a word is* $p = \Theta(\log N)$ *bits, represented by an element of* $\mathbb{Z}_{2^p}$*) by* $P = \Theta(N^{1+\gamma-\varepsilon})$ *machines, each with local memory* $s = O(N^\varepsilon)$ *words:*

1. *Initially, the input is arbitrarily distributed across the first* $\lceil \frac{N}{s} \rceil$ *machines.*

2. *In each round, each machine prepares, as an arbitrary function of its local memory, messages to send to other machines. The total size of messages prepared by any machine is at most* $s$ *words.*

3. *At the end of each round, the messages are placed in the local memory of the intended recipients. The protocol ensures that the messages received by any machine has total size at most* $s$ *words.*

4. *After the* $R$*-th round, the output is stored in the memory of the first* $\lceil \frac{N}{s} \rceil$ *machines.*

*We say an MPC protocol* $\pi$ *computes a function* $f : \mathbb{Z}_{2^p}^N \to \mathbb{Z}_{2^p}^N$, *if for any* $X \in \mathbb{Z}_{2^p}^N$, $\pi(X) = f(X)$, *where* $\pi(X)$ *is the output of* $\pi$ *with the input* $X$.

The primary measure of complexity considered in this paper is the number of rounds $R$. The round complexity of numerous classical algorithmic problems is well understood. For example, there are simple MPC protocols for graph connectivity ($O(\log N)$ rounds) and sorting ($O(1)$ rounds) [20].

---

[1] Such assumptions are necessary for establishing the equivalence with the MPC model, since the MPC model allows arbitrary computation on the local memory for each machine. That said, many concrete MPC algorithms do have a simple local algorithm which can be simulated by a small MLP.

# 3 Approximate Nearest Neighbor Attention

In this section, we introduce Approximate Nearest Neighbor Attention (ANNA), an attention mechanism inspired by the approximate nearest neighbor (ANN) search problem. We first outline the approximate nearest neighbor search problem and present locality-sensitive hashing (LSH), a core technique for ANN (Section 3.1). We then formally define ANNA and provide a sub-quadratic time algorithm based on LSH for computing ANNA with theoretical guarantees (Section 3.2).

## 3.1 Approximate nearest neighbor and locality sensitive hashing

We first define the Approximate Nearest Neighbor (ANN) search problem.

**Definition 3.1** (ANN search problem [30, 38]). *Given a dataset $D$ of $N$ points lying in a metric space $Y$ and parameters $c, r > 0$, build a data structure that, given a query $q \in Y$ within distance at most $r$ from $D$, returns any point in $D$ that is within distance $cr$ from $q$.*

In the modern machine learning setting, we want to develop fast algorithms for ANN search in the high-dimensional metric space with sub-linear query time. A well-known tool that achieves this runtime and provable approximation guarantees is locality sensitive hashing (LSH). For simplicity, we assume the metric space is $m$-dimensional Euclidean space.

**Definition 3.2** (Locality Sensitive Hashing [30]). *Fix a parameter $r > 0$, an approximation factor $c > 1$ and a set $U$. Then a family $\mathcal{H}$ of hash functions $h : \mathbb{R}^m \to U$ is $(r, cr, p_1, p_2)$-sensitive if the following holds for any $x, y \in \mathbb{R}^m$:*

- *if $\|x - y\| \leq r$, then $\Pr_{h \in \mathcal{H}}[h(x) = h(y)] \geq p_1$, and*
- *if $\|x - y\| > cr$, then $\Pr_{h \in \mathcal{H}}[h(x) = h(y)] \leq p_2$.*

*The family $\mathcal{H}$ is called an LSH family with quality $\rho = \frac{\log(1/p_1)}{\log(1/p_2)}$.*

A typical LSH-based algorithm can solve the ANN search problem with space $O(N^{1+\rho})$ and query time $O(N^\rho)$, and $\rho$ can be as small as $1/c^2$ [4].

## 3.2 Transformer based on Approximate Nearest Neighbor Attention

We first define a family of models where only tokens with sufficiently "neighborly" queries and keys attend to one another and then provide an efficient implementation of a subset of this class. ANNA attention units treat attention query vectors as queries to approximate nearest neighbor search and key vectors as data points. ANNA retrieves and weights value vectors according to approximate nearest neighbor thresholds. The following definition formalizes this family of models.

**Definition 3.3** (ANN Attention). *An Approximate Nearest Neighbor Attention (ANNA) mechanism $\mathrm{ANNA}_{Q,K,V}$ with query, key, and value embedding functions $Q, K, V : \mathbb{R}^d \to \mathbb{R}^m$ and (non-negative) parameters $r, c, \ell, \eta$, is a (possibly randomized) mechanism that performs the following computation on an input $X \in \mathbb{R}^{N \times d}$:*

$$\mathrm{ANNA}_{Q,K,V}(X)_i := \sum_{j=1}^{N} w_{i,j} v_j \quad \text{for all } i \in [N],$$

*for some non-negative weights $w_{i,j} \geq 0$ with $\sum_j w_{i,j} = 1$ that satisfy the following. With probability at least $1 - \eta$, for all $i \in [N]$,*

- $w_{i,j} > 0 \Rightarrow k_j \in \mathcal{N}(q_i, cr)$
- $k_j \in \mathcal{N}(q_i, r) \Rightarrow w_{i,j} \geq \frac{1}{(|\mathcal{N}(q_i, cr)| - 1)\ell + 1}$

*where $q_i := Q(X)_i$, $k_i := K(X)_i$, $v_i := V(X)_i$, and $\mathcal{N}(q, t) := \{k \in \{k_j\}_{j=1}^{N} : \|q - k\| \leq t\}$.*

*We define* ANNA layers *and* ANNA transformers *in a completely analogous fashion as (standard) attention layers and transformers are defined (Definitions 2.1 and 2.2).*

**Algorithm 1** ANNA implementation with LSH family $\mathcal{H}$, $\ell$ hash tables, and $z$ hash functions/table

---

**Input:** Input $X \in \mathbb{R}^{N \times d}$
**Output:** ANNA output for each of the query.
1: Let $q_i = Q(X)_i$, $k_i = K(X)_i$, and $v_i = V(X)_i$ for all $i \in [N]$.
2: **for** $u = 1$ to $\ell$ **do**                                                                     ▷ **Preprocessing phase**
3:     Sample $z$ hash functions $h_{u,1}, h_{u,2}, \ldots, h_{u,z}$ i.i.d. from $\mathcal{H}$.
4:     Create empty hash table $T_u$ indexed by hash codes (below).
5:     **for** each key-value pair $(k_j, v_j)$ **do**
6:         Compute hash code $g_u(k_j) = (h_{u,1}(k_j), h_{u,2}(k_j), \ldots, h_{u,z}(k_j))$.
7:         **if** $T_u[g_u(k_j)]$ is empty, **then** $T_u[g_u(k_j)] := (v_j, 1)$ **else** $T_u[g_u(k_j)] \mathrel{+}= (v_j, 1)$.
8: Initialize a dictionary **attn** $\leftarrow \{(q_1, 0), (q_2, 0), \ldots, (q_N, 0)\}$
9: **for** each query $q_i$ **do**                                                                        ▷ **Query phase**
10:     $v_{\text{sum}} \leftarrow 0; \text{count} \leftarrow 0$
11:     **for** $u = 1$ to $\ell$ **do**
12:         Compute hash code $g_u(q_i) = (h_{u,1}(q_i), \ldots, h_{u,z}(q_i))$
13:         **if** $T_u[g_u(q_i)] = (v, a)$ is not empty, **then** $v_{\text{sum}} \mathrel{+}= v$ and count $\mathrel{+}= a$
14:     **attn**$[q_i] \leftarrow v_{\text{sum}}/\text{count}$
15: **return attn**

---

The parameters $r, c$ have the same semantics as in ANN search. The parameter $\ell$ captures how much "attention weight" is spread over keys that are not $r$-near neighbors of a query. The failure probability $\eta$ allows for randomization, which is typical of ANN search algorithms like LSH.

The above definition represents a set of constraints that all ANNA units must satisfy rather than a specific algorithmic implementation. As a result, a wide variety of models satisfy the definition, including softmax attention with bounded query and key vectors, and *Exact-Match Attention (EMA)* where $w_{i,j} > 0$ if and only if $q_i = k_j$. Not all such models admit computationally efficient implementations. To identify sub-quadratic ANNA models, we present an LSH-based implementation of ANNA that computes satisfying weight vectors $w_{i,j}$ for specific choices of parameters $r, c, \ell, \eta$.

We define a hash function family $G = \{g : p \in \mathbb{R}^d \mapsto (h_1(p), h_2(p), \ldots, h_z(p)) \in U^z \mid h_i \in \mathcal{H}, \forall i \in [z]\}$ and sample $\ell$ hash functions, $g_1, \ldots, g_l$, from $G$ independently and uniformly at random, giving $\ell$ hash tables. Each hash code corresponds to a hash bucket in the hash table and each hash bucket maintains a sum of $v$'s and count of $k$'s that falls into this bucket. We preprocess all the key, value pairs by storing them in the hash tables. For each $(k_i, v_i)$, $i \in [N]$, compute the hash codes of $k_i$, $g_1(k_i), g_2(k_i), \ldots, g_\ell(k_i)$, and update the sum and count for the buckets corresponding to $g_1(k_i), g_2(k_i), \ldots, g_\ell(k_i)$ respectively. For each query $q_i, i \in [N]$, search and retrieve all the values and counts from $g_1(q_i), g_2(q_i), \ldots, g_\ell(q_i)$. Then compute the averaged value by summing up all the values in the $\ell$ buckets, divided by the sum of counts. See Algorithm 1 for the details.

**Theorem 3.4** (LSH algorithm guarantee for ANNA). *Fix $c > \sqrt{3}$, LSH family $\mathcal{H}$ that is $(r, cr, p_1, p_2)$-sensitive with quality $\rho < 1/3$, $\ell = \Theta(N^{3\rho} \log N)$, and $z = \Theta(\log_{1/p_2} N)$. Then Algorithm 1 (with $\mathcal{H}$, $\ell$, and $z$) implements an ANNA mechanism with parameters $r, c, \ell$ and $\eta = O(1/N^{1-3\rho})$.*

We leave the full proof of Theorem 3.4 to Appendix A. The total runtime of Algorithm 1 is $O(mN^{1+3\rho} \log_{1/p_2} N)$, assuming sampling from the LSH family and evaluating a hash function requires $O(m)$ time and the numerical inputs to Algorithm 1 are specified with $p = \Theta(\log N)$ bits of precision. The total space used is $\tilde{O}(mN^{1+3\rho})$ bits.

**Remark 3.5.** *The memory complexity of Algorithm 1 can be further improved to $\tilde{O}(mN)$ bits with the same time complexity by storing only one hash table with each entry keeping track of the value for each query. See Algorithm 2 for the detailed implementation in Appendix A.*

**Remark 3.6.** *The weight $w_{i,j}$ depends on the number of hash collisions between $q_i$ and $k_j$, and is typically a function of the distance $\Delta := \|q_i - k_j\|$. For example, if we use the random hyperplane LSH from [6], then $w_{i,j} \propto \exp(-\Delta^2 \log(m)/(4 - \Delta^2))$.*

In the remainder of this paper, our ANNA-based constructions are meant to refer to their efficient implementation by Algorithm 1 with a suitable choice of $r$ and arbitrarily large $c$.

# 4 Efficient transformers and MPC

We prove a sharp equivalence between ANNA-transformer and MPC in the regime of sub-linear local memory and sub-quadratic number of machines (Sections 4.1 and 4.2). We also show that ANNA subsumes alternative low-rank sub-quadratic attention mechanisms. (Section 4.3).

## 4.1 ANNA-transformer can simulate MPC

The following theorem shows that any $R$-round MPC protocol with sub-linear local memory can be simulated by an ANNA-transformer with $O(R)$ layers and sub-linear number of heads and embedding dimension. The full proof of Theorem 4.1 is in Appendix B.

**Theorem 4.1** (ANNA simulates MPC). *Fix constants $0 < \varepsilon < \varepsilon' < 1$. For any deterministic $R$-round $(\varepsilon, \varepsilon)$-MPC protocol $\pi$, there exists an ANNA-transformer $T$ with $L = O(R)$ layers, $H = O(N^{(\varepsilon'-\varepsilon)/4})$ heads per layer, and embedding dimension $m = O(N^{\varepsilon'})$, such that $T(\texttt{input}) = \pi(\texttt{input})$ for all $\texttt{input} \in \mathbb{Z}_{2^p}^N$.*

*Proof sketch.* In fact, we show that the special case of ANNA-transformer whose approximation factor $c \to \infty$ and $r = 0$ is already suffice to simulate MPC. In such case, ANNA for each query is equivalent to finding only the keys that exactly match the query; we call this Exact-Match Attention (EMA), and formally define it in Appendix B.

In our simulation, we treat each input token as the local machine, and all the local computation is handled by the element-wise functions $Q, K, V$. The bulk of the proof is to handle the message delivery between machines using EMA. By Proposition 24 of [53], we can assume that each machine only sends messages to at most $\alpha = O(N^\delta)$ machines for some $\delta < \varepsilon$. We assign a unique positional encoding or identifier to each machine, and this encoding serves as a unique key to retrieve the message in each machine. The high level idea is to create a query for each machine and a key for each destination machine and the associated value is the embedding of the message sent to the destination machine in the protocol. Since each machine can send at most $\alpha$ messages to other machines, we create $\alpha$ EMA heads and each head is responsible for one outgoing message for all the $N$ machines. Each machine retrieves the message sent to them by having a query in each head. Since the messages are averaged together, we use the same embedding mechanism from Lemma 3.2 of [55] to allow error correction in the element-wise operations. $\qquad\square$

This gives us a sub-quadratic time reduction from MPC to ANNA-transformer: i.e., the communication process can be implemented in near linear time, whereas it is quadratic for standard attention. In addition, this ties ANNA-transformer in the existing MPC hierarchy [53]: any problem solvable by an $R$-round, $O(N^\varepsilon)$-memory MPC protocol can be solved by $O(R)$-layer ANNA-transformer with $mH = O(N^{\varepsilon+\delta})$, for some $\delta > 0$. For example, following Theorem 3.1 of [26], $O(1)$-layer ANNA-transformer can solve 3-SUM with $mH = O(N^{1/2+\delta})$.

## 4.2 MPC can simulate ANNA-transformer

The following theorem (proved in Appendix C) shows that any $L$-layer ANNA-transformer (as implemented by Algorithm 1) can be simulated by a $O(L)$-round MPC protocol. Since Algorithm 1 is randomized, it uses a random seed to sample the hash functions from the LSH family. The simulation assumes access to the random seeds needed for all layers in the ANNA-transformer.

**Theorem 4.2** (MPC simulates ANNA). *Fix constants $0 < \varepsilon < \varepsilon' < 1$. For any $L$-layer ANNA-transformer $T$ (as implemented by Algorithm 1) with $mH = O(N^\varepsilon)$, there exists a $O(L/(\varepsilon' - \varepsilon))$-round MPC protocol $\pi$ with local memory $s = O(N^{\varepsilon'})$ and $P = O(N^{1+\varepsilon-\varepsilon'+3/c^2})$ machines such that $\pi(\texttt{input}) = T(\texttt{input})$ for all $\texttt{input} \in \mathbb{Z}_{2^p}^N$.*

Observe that the number of machines used in the simulation of the ANNA-transformer can be strongly sub-quadratic (and in fact, close to linear when $c$ is large). In contrast, the simulation of a standard transformer from [55] requires $N^2$ machines. As previously discussed (Section 1.2), this shows that ANNA-transformer more sharply characterizes efficient MPC protocols than standard transformers do. On the other hand, by Theorem 4.2, round-complexity lower bounds for MPC directly imply depth lower bounds for ANNA-transformers. This argument was used in [55] to establish (conditional)

depth lower bounds for standard transformers on problems such as graph connectivity and $k$-hop induction heads; these lower bounds also hold for ANNA-transformers.

### 4.3 ANNA-transformer can simulate low-rank transformers

As mentioned in Section 1, there are many proposals for efficient attention alternatives. In this section, we focus on the sub-quadratic alternatives based on low-rank approximations of the attention matrix. Specifically, we ask the following: *what problems are intrinsically easy for ANNA but hard for low-rank approximation attention, and vice versa?*

**Definition 4.3** (Low-rank attention)**.** *A low-rank attention is specified by two feature maps $Q', K' : \mathbb{R}^d \to \mathbb{R}^r$ for some $r \ll N$ (with the intention of approximating $\mathrm{softmax}(Q(X)K(X)^\intercal)$ by $Q'(X)K'(X)^\intercal$). On input $X \in \mathbb{R}^{N \times d}$, it computes $Q'(X)K'(X)^\intercal V(X)$ by first computing $K'(X)^\intercal V(X) \in \mathbb{R}^{r \times m}$, and then left-multiplying by $Q'(X)$.*

Note that [55] gives a lower bound of any low-rank attention for the $k$-hop problem. Later in Section 5, we give a construction of $O(\log k)$-depth ANNA-transformer solving $k$-hop, and this directly gives a type of problem that is easy for ANNA but hard for low-rank attention. However, is there any problem that is easy for low-rank attention but hard for ANNA?

The following theorem answers the question by showing any $L$-layer low-rank attention-transformer can be simulated by a $O(L)$-layer ANNA-transformer. So, under the time and parameter-efficient regime (sub-linear rank and embedding dimension), low-rank attention-transformer is no stronger than ANNA-transformer.

**Theorem 4.4** (ANNA simulates low-rank attention)**.** *For constants $0 < \varepsilon < \varepsilon' < 1$, any low-rank attention based transformer with depth $L$, rank $r$, embedding dimension $m$ and $rm = O(N^\varepsilon)$ can be simulated by an ANNA-transformer with depth $O(L)$, number of heads $H = O(N^{(\varepsilon'-\varepsilon)/4})$ and embedding dimension $m = O(N^{\varepsilon'})$.*

We prove Theorem 4.4 by first using $O(L/(\varepsilon' - \varepsilon))$-round MPC to simulate $L$-layer low-rank transformer, and then Theorem 4.1 give us the simulation of $L$-layer low-rank transformer with ANNA-transformer through MPC. The full proof is given in Appendix D.

**Other efficient attention mechanisms based on nearest neighbor search.**   Reformer [35] is another efficient attention based on LSH. In Reformer, the input tokens are sorted by their (scalar) hash values. Then, this sorted list is split into equal-sized chunks, each containing only $O(1)$-many tokens. Standard attention is applied within each chunk. We show that the expressive power of Reformer must come from the sorting operation: without sorting, the restriction of attention within each constant-size chunk prevents Reformer from even computing basic functions like "average" with $O(1)$ layers (regardless of the embedding dimension); details are given in Appendix E.

KDEformer [64] and HyperAttention [27] approximate softmax attention matrices as a sum of a sparse matrix and a low-rank matrix. They use LSH techniques to find sparse elements (i.e., heavy elements in the attention matrix) and low-rank attention for the remaining components. Theorem 4.4 indicates this low-rank part does not substantially increase the representational power.

## 5   ANNA-transformer for reasoning tasks

In this section, we study ANNA-transformer on two concrete reasoning tasks: Match2 [52] and $k$-hop [55]. These tasks are benchmarks for evaluating the reasoning capabilities of transformers, and they separate different neural architectures in terms of their representational strengths.

### 5.1   ANNA-transformer solves Match2

The Match2 task [52] measures the ability of a model to associate paired elements with one another. We show that a single ANNA mechanism can solve Match2.

**Definition 5.1** (Match2)**.** *Given an input sequence $X = (x_1, \ldots, x_N) \in [M]^N$ for some $M \leq \mathrm{poly}(N)$, the $i$-th output of $\mathrm{Match2}(X)$ is $\mathbb{1}\{\exists j \cdot x_i + x_j = 0 \bmod M\}$ for all $i \in [N]$.*

**Theorem 5.2.** *For any $N, M = N^{O(1)}$, there exists an ANNA-transformer $T$ with one layer, one attention head, and embedding dimension 1 such that $T(X) = \mathrm{Match2}(X)$ for all $X \in [M]^N$.*

## 5.2  ANNA-transformer solves $k$-hop

The induction heads (a.k.a. associative recall) task [18] is a reasoning task that predicts the next token by completing the most recent bigram. It has been identified as an important mechanism for the emergent "in-context learning" ability of LLMs.

**Definition 5.3** (Induction heads). *Let $\Sigma$ be a finite alphabet and $w \in \Sigma^N$. For each $i \in [N]$, define*

$$\sigma(w, i) = \max\{\{0\} \cup \{j \in \mathbb{N} : j \le i, w_{j-1} = w_i\}\}.$$

*The* induction head *task is to compute, for each $1 \le i \le N$, the value of $w_{\sigma(w,i)}$.*

For example, let $\Sigma = \{\mathtt{a}, \mathtt{b}, \mathtt{c}\}$ and $w = \mathtt{aabcbabca}$. Then $w_{\sigma(w,9)} = \mathtt{b}$ because the 9th token is $\mathtt{a}$, and the last occurrence of $\mathtt{a}$ before position 9 (which is in position 6) is followed by $\mathtt{b}$.

The following theorem shows that our ANNA-transformer can solve induction heads problem using constant number of layers and sub-linear embedding dimension and number of heads.

**Theorem 5.4.** *Fix constants $0 < \varepsilon < \varepsilon' < 1$. There exists an ANNA-transformer $T$ with $L = O(1)$ layers, $H = O(N^{(\varepsilon'-\varepsilon)/4})$ heads per layer, and embedding dimension $m = O(N^{\varepsilon'})$ such that $T(w)_i = w_{\sigma(w,i)}, \forall i \in [N]$, for all $w \in \Sigma^N$.*

We prove Theorem 5.4 (in Appendix F.2) by constructing a constant-round MPC algorithm for induction heads, and then applying Theorem 4.1 to convert it into an ANNA-transformer.

The induction heads task was generalized by [55] to a $k$-step variant called "$k$-hop".

**Definition 5.5** ($k$-hop induction heads). *Let $\Sigma$ be a finite alphabet and $w \in \Sigma^N$. Let $\sigma^k(w, i)$ denote a $k$-fold composition of $\sigma(w, \cdot)$ from the previous definition. The $k$-hop induction head task is to compute $w_{\sigma^k(w,i)}$ for each $1 \le i \le N$.*

Using the same example where $\Sigma = \{\mathtt{a}, \mathtt{b}, \mathtt{c}\}$, $w = \mathtt{aabcbabca}$ and $k = 2$, we have $w_{\sigma(\sigma(w,9))} = \mathtt{a}$ because the last occurrence of $\mathtt{b}$ before position 7 is followed by $\mathtt{a}$.

As was done in [55], we construct a $O(\log k)$-round MPC algorithm for $k$-hop using function composition, thus yielding a logarithmic depth scaling for ANNA-transformers on this task.

**Theorem 5.6.** *Fix constants $0 < \varepsilon < \varepsilon' < 1$, any $k \in \mathbb{N}$ and alphabet $\Sigma$ with $|\Sigma| = O(N)$. There exists an ANNA-transformer $T$ with $L = O(\log k)$ layers, $H = O(N^{(\varepsilon'-\varepsilon)/4})$ heads per layer, and embedding dimension $m = O(N^{\varepsilon'})$ such that $T(w)_i = w_{\sigma^k(w,i)}, \forall i \in [N]$, for all $w \in \Sigma^N$.*

The full construction is given in Appendix F.2. We note that while prior works have given transformer constructions for $k$-hop [12, 55] (and Match2 [52]), these results do not directly imply ANNA-transformers constructions, given the differences in the architectures.

Prior work [55] showed that multi-layer recurrent nets and low-rank sub-quadratic attention [16, 32] are unable to solve $k$-hop unless the depth is $\Omega(k)$ or their memory size/embedding dimension is $\Omega(N/k^6)$. On contrast, ANNA-transformer achieves both $O(\log k)$-depth and sublinear (in $N$) width. In this sense, the $k$-hop task separates ANNA-transformer from these other efficient neural architectures.

## 5.3  Experiments on Match2 and induction heads

We empirically test the performance of ANNA-transformer on the Match2 and induction heads tasks. Experimental details are given in Appendix G. Since Algorithm 1 is not differentiable, we train a softmax version of attention as a surrogate, and then distill from the trained model to an ANNA-transformer (based on Algorithm 1 with angular LSH [6]). Our softmax attention normalizes all the queries and keys in $Q(X)$ and $K(X)$ to have unit norm, and computes $\mathrm{softmax}(\beta \cdot Q(X)K(X)^\intercal)V(X)$ with a tunable temperature parameter $\beta > 0$.

The Match2 dataset is generated the same way as [37] with context length $N = 32$ and upper bound $M = 37$. One-layer ANNA-transformers are able to achieve zero error with $\ell = 8$ hash tables and $z = 1$ hash function per table. See Figure 1a for the detailed performance. For induction heads, we use the dataset from [55] with number of hops $k = 1$, context length $N = 100$ and alphabet size $|\Sigma| = 4$. A two-layer ANNA-transformer achieved highly nontrivial error with $\ell = 32$ hash tables and $z = 2$ hash functions per table; with more hash tables, the error rate was as low as $0.1$. See Figure 1b for the detailed results.

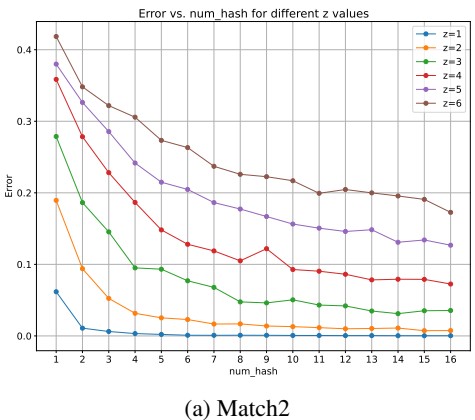

(a) Match2

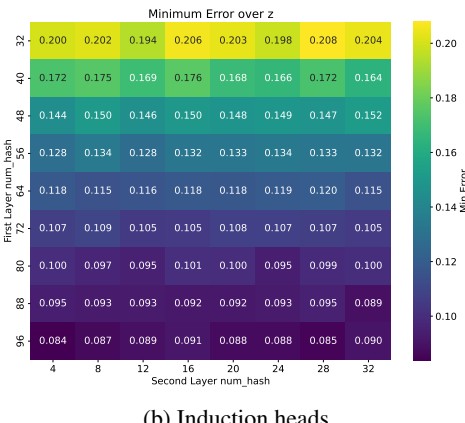

(b) Induction heads

Figure 1: All errors are averaged over 10 runs. (a) Error rate on Match2: x-axis denotes the number of hash tables $\ell$, and different colors correspond to different numbers $z$ of hash functions per hash table. (b) Error rate on induction heads: Rows correspond to the number of hash tables in the first layer, columns correspond to the number of hash tables in the second layer. The reported error rate is the best achieved over the choice of $z \in \{1, 2, 3, 4\}$.

## 6  Conclusion and future work

In this work, we propose a more efficient class of neural architecture, the ANNA-transformers, which not only preserve the representational power of standard transformer characterized by the MPC framework but also yield a tighter equivalence with the MPC model. Furthermore, we show that constant layers of ANNA-transformers can simulate constant layers of low-rank transformer, and can solve reasoning tasks such as Match2 and $k$-hop tasks in near-optimal depth.

There are some interesting directions we leave for future work. While our ad hoc training method was effective as a proof-of-concept, it is desirable to develop a principled training method that directly optimizes the performance of an ANNA-transformer (or a differentiable variant thereof), rather than that of a surrogate model. Also, our empirical validation was limited to small synthetic datasets; extending these experiments to large-scale, real-world benchmarks is a promising next step.

## 7  Acknowledgements

Part of this work was done at the "Modern Paradigms in Generalization" and "Special Year on Large Language Models and Transformers, Part 1" programs at the Simons Institute for the Theory of Computing, Berkeley in 2024. We acknowledge support from the ONR under grants N00014-24-1-2700, N00014-22-1-2713, from the NSF under grant CCF2008733, and an award from the Columbia Center of AI Technology in collaboration with Amazon.

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

# A  Proof of Theorem 3.4

We restate Theorem 3.4 here, which gives the theoretical guarantee for Algorithm 1.

**Theorem A.1** (LSH algorithm guarantee for ANNA; Theorem 3.4). *Fix $c > \sqrt{3}$, LSH family $\mathcal{H}$ that is $(r, cr, p_1, p_2)$-sensitive with quality $\rho < 1/3$, $\ell = \Theta(N^{3\rho} \log N)$, and $z = \Theta(\log_{1/p_2} N)$. Then Algorithm 1 (with $\mathcal{H}$, $\ell$, and $z$) implements an ANNA mechanism with parameters $r, c, \ell$ and $\eta = O(1/N^{1-3\rho})$.*[2]

*Proof.* Our algorithm applies for the regime with large approximation factor, i.e., $c > \sqrt{3}$. Since we only want the nearest neighbors within distance $cr$ with the query point, we want to bound the probability of two points with distance greater than $cr$ to fall into the same bucket. Consider the family $G$ with $\Pr_{g \in G}[g(x) = g(y)] \leq \frac{0.1}{N^3}$, if $\|x - y\| > cr$. Then for each bucket, the expected number of collision ($x, y$ fall into the same bucket and $\|x-y\| > cr$) is less than $N \cdot \Pr_{g \in G}[g(x) = g(y)] \leq \frac{0.1}{N^2}$. Therefore, by Markov's inequality, for each bucket, with probability greater than $1 - \frac{0.1}{N^2}$, there is no collision within the bucket. Then, by union bound over all the non-empty bucket (there are at most $N$ of them), with probablity greater than $1 - \frac{0.1}{N}$, there is no collision in one hash table. By [30], $z = O(\log_{1/p_2} N)$, i.e., each hash function $g \in G$ is composed of $O(\log_{1/p_2} N)$ hash functions sampled from the LSH family $\mathcal{H}$, which suffices to achieve $\Pr_{g \in G}[g(x) = g(y)] \leq \frac{0.1}{N^3}$ whenever $\|x - y\| > cr$.

On the other hand, since the probability of collision is very small, the success probability (when $\|x - y\| \leq r$), namely $p = \Pr_{g \in G}[g(x) = g(y)] = N^{-3\rho}$ (recall that $\rho = \frac{\log 1/p_1}{\log 1/p_2}$), is also somewhat small. However, we can boost the success probability by using multiple hash tables. Let $\ell$ denote the number of hash tables. Then for each $q_i$, the probability of its $r$-nearest neighbor $k$ ($k \in \mathcal{N}(q_i, r)$) falls into different bucket with $q_i$ for all $\ell$ tables is upper bounded by $(1 - p)^\ell$. By union bound over all possible nearest neighbors and all $q_i$'s, the failure probability is bounded by $N^2(1-p)^\ell$. Assume we want the failure probability to be less than some $\delta > 0$, then we want $N^2(1-p)^\ell \leq \delta$. Taking logarithm of both sides, and using a Taylor expansion of $\log(1-x)$ for sufficiently small $x$, we find that $\ell = O(N^{3\rho}(\log N + \log 1/\delta))$ suffices for success probability $1 - \delta$.

Therefore, by union bound over all $\ell$ hash tables, with probability $1 - \frac{0.1}{N^{1-3\rho}}$, there is no collision in all the hash tables, which implies $w_{i,j} = 0$ if $\|k_j - q_i\| > cr$. By setting $\delta = \frac{0.1}{N^{1-3\rho}}$, we get $\ell = O(N^{3\rho} \log N)$. Hence, the total failure probability $\eta$ is bounded by $\delta + \frac{0.1}{N^{1-3\rho}}$ which is $O(1/N^{1-3\rho})$.

If $\|k_j - q_i\| \leq r$, from the guarantee above, we know that $k_j$ collides with $q_i$ at least once in the $\ell$ hash bucket. This implies $w_{i,j} \geq 1/count$, where count is the number of all the collisions in the $\ell$ hash buckets that $q_i$ retrieves. In the worst case, all the $k \in \mathcal{N}(q_i)$ collides with $q_i$ in all $\ell$ hash tables except for $k_j$ only colliding once. Therefore, count $\leq (\mathcal{N}(q_i) - 1) \cdot \ell$, and this gives us $w_{i,j} \geq \frac{1}{(\mathcal{N}(q_i)-1)\cdot\ell+1}$. $\qquad\square$

**Runtime and memory usage.** One can see that for each query, we need to evaluate $O(N^{3\rho} \log_{1/p_2} N)$ hash functions and compute sum of $m$-dimensional vectors, so the total runtime is $O(mN^{1+3\rho} \log_{1/p_2} N)$. During the preprocessing, we need to store $N^{3\rho}$ hash tables and the sum of values, each with at most $N$ buckets, so the total memory is $O(mN^{1+3\rho} \log N)$ bits. In fact, the space used can be further improved to $O(mN \log N)$ bits. Instead of maintaining $\ell$ hash tables, one can just store 1 hash table of size $O(mN \log N)$ with each entry responsible for tracking the values for each query. For each round of hashing ($\ell$ rounds in total), hash all queries using the hash functions and creates empty buckets for them. Then, hash each key, and if the key hashes to an existing query bucket, its value is added (along with a count). After processing keys, each query accumulates the values and counts from its corresponding bucket. We give the memory-efficient implementation in Algorithm 2.

---

[2]Optimal (data-oblivious) LSH schemes achieve $\rho = 1/c^2 + o(1)$ [4]. Since we assume $c > \sqrt{3}$, the failure probability $1/\text{poly}(N)$ decreases to zero with $N$.

**Algorithm 2** Linear memory ANNA implementation with LSH family $\mathcal{H}$, $\ell$ hash tables, and $z$ hash functions/table

**Input:** Input $X \in \mathbb{R}^{N \times d}$
**Output:** ANNA output for each of the query.
 1: Let $q_i = Q(X)_i$, $k_i = K(X)_i$, and $v_i = V(X)_i$ for all $i \in [N]$.
 2: Initialize an array of tuples $A$, and $A[i] \leftarrow (0,0), \forall i \in [N]$.
 3: **for** $u = 1$ to $\ell$ **do**
 4:      Sample $z$ hash functions $h_{u,1}, h_{u,2}, \ldots, h_{u,z}$ i.i.d. from $\mathcal{H}$.
 5:      Create empty hash table $T_u$ indexed by hash codes of queries (below).
 6:      **for** each query $q_i$ **do**
 7:          Compute hash code $g_u(q_i) = (h_{u,1}(q_i), \ldots, h_{u,z}(q_i))$.
 8:          Create an entry in $T_u$ indexed by $g_u(q_i)$ and $T_u[g_u(q_i)] \leftarrow (0,0)$.
 9:      **for** each key-value pair $(k_j, v_j)$ **do**
10:          Compute hash code $g_u(k_j) = (h_{u,1}(k_j), h_{u,2}(k_j), \ldots, h_{u,z}(k_j))$.
11:          **if** $T_u[g_u(k_j)]$ exists in $T_u$, **then** $T_u[g_u(k_j)] \mathrel{+}= (v_j, 1)$.
12:      **for** each query $q_i$ **do**
13:          $A[i] \mathrel{+}= T_u[g_u(q_i)]$
14: Initialize a dictionary **attn** $\leftarrow \{(q_1, 0), (q_2, 0), \ldots, (q_N, 0)\}$.
15: **for** each query $q_i$ **do**
16:      **attn** $\leftarrow A[i][0]/A[i][1]$
17: **return attn**

# B   ANNA-transformer can simulate MPC

Our simulation of MPC using ANNA-transformers uses only a special case of ANNA, which we call *Exact Match Attention (EMA)*. In EMA, we require the key to be *exactly the same as* the query for it to be considered in the attention matrix. We show that this special case already suffices to simulate MPC.

**Definition B.1** (EM Attention). *Let $X \in \mathbb{R}^{N \times d}$ be the input embedding, $Q, K, V : \mathbb{R}^{N \times d} \to \mathbb{R}^{N \times d}$ be query/key/value embedding functions. For any query $q$, let $\mathcal{N}(q) = \{k_j \in K : k_j = q\}$. For each query $q_i$, the* Exact Match attention *computes*

$$\text{EMA}_{K,V}(q_i) = \begin{cases} \frac{1}{|\mathcal{N}(q_i)|} \displaystyle\sum_{j \in \mathcal{N}(q_i)} v_j & \text{if } \mathcal{N}(q_i) \neq \emptyset \\ \mathbf{0} & \text{otherwise.} \end{cases}$$

EMA layer and EMA-transformer are defined analogously. To see that EMA is a special case of ANNA, notice that in ANNA, we can set $r = 0, c \to \infty$ and $w_{i,j} = \frac{1}{|\mathcal{N}(q_i)|}$ such that it becomes exactly the same as EM attention. EMA also admits a near linear-time algorithm: sort all the keys first (using a lexicographic ordering) in time $O(dN \log N)$ and space $O(dN)$; at query time, perform binary search in time $O(d \log N)$ per query.

We first give a simulation that directly simulates the $R$-round $(\varepsilon, \varepsilon)$-MPC using $L = R + 1$ layers but large embedding dimensions to showcase the core idea of the proof.

**Theorem B.2** (EMA simulates MPC). *For constant $0 < \varepsilon < 1$, any deterministic $R$-round MPC protocol $\pi$ with $N$ machines with $s = O(N^{\varepsilon})$ words local memory, there exists an EMA-transformer $T$ with depth $L = R + 1$, number of heads $H = O(N^{\varepsilon})$, and embedding dimension $m = O(N^{5\varepsilon} \log N)$, such that $T(\text{input}) = \pi(\text{input})$ for all $\text{input} \in \mathbb{Z}_{2^p}^N$.*

*Proof.* For any $R$-round MPC protocol $\pi$ with $N$ machines that maps the input to output, we define the intermediate steps for local computation phase and message transimission phase. We denote the input to all the machines before the local computation as $\text{MachineIn}_1, \text{MachineIn}_2, \ldots, \text{MachineIn}_R$, and denote the information after deterministic local computations $(\text{Local}_r^i)_{r \in [R], i \in [N]}$ as $\text{MachineOut}_1, \text{MachineOut}_2, \ldots, \text{MachineOut}_R$, where $\text{MachineOut}_r^i = \text{Local}_r^i(\text{MachineIn}_r^i)$. In the communication (message transimission) phase, we need to route the messages to the correct machines ie from $\text{MachineOut}_r$ to $\text{MachineIn}_{r+1}$.

In our simulation, each token input to the EMA-transformer plays the role of a machine in the MPC protocol. We simulate the local computation functions $(\texttt{Local}_r^i)_{r \in [R], i \in [N]}$ by the element-wise functions $Q(\cdot), K(\cdot), V(\cdot)$ in the architecture. Therefore, the simulation process can be partitioned into 3 different parts:

1. Initialization. The `input` feed into EMA-transformer is distributed in the $N$ tokens, and we need to transfer than into the first $\lceil \frac{N}{s} \rceil$ tokens/machines to match $\texttt{MachineIn}_1$.

2. Routing (message transmission). After the local computation in each round $r$, we need to communicate the messages from $\texttt{MachineOut}_r$ to $\texttt{MachineIn}_{r+1}$.

3. Final output. The MPC `output` is distributed in the first $\lceil \frac{N}{s} \rceil$ tokens/machines, and we need to distributed them back to the N tokens.

The following 3 lemmas construct the elements for each of these 3 parts.

We first show the message transmission part of MPC can be simulate by the EMA-transformer. Recall that after $r$ rounds of local computation, each machine $i$ has a set of messages it wants to send to other machines, denoted by $\texttt{MachineOut}_r^i = \{(\texttt{Msg}_{\texttt{dest}}^i, \texttt{dest}) : \texttt{dest} \in \texttt{sent}^i\}$, where $\texttt{sent}^i$ is the set of machine indices that machine $i$ will sent the message to and $\texttt{Msg}_{\texttt{dest}}^i$ is the message machine $i$ send to machine $\texttt{dest}$. After the message communication phase, each machine $i$ has the set of messages it receives from other machines, denoted by $\texttt{MachineIn}_{r+1}^i = \{(\texttt{Msg}, \texttt{Src}) : (\texttt{Msg}, \texttt{i}) \in \texttt{MachineOut}_r^{\texttt{Src}}\}$. Since each machine can only send/receive $s$ words, we have $\sum_{\texttt{dest} \in \texttt{sent}^i} |\texttt{Msg}| \leq$ s and $\sum_{(\texttt{Msg},\texttt{i}) \in \texttt{MachineOut}_r^{\texttt{Src}}} |\texttt{Msg}| \leq$ s for all machine $i$. We call this process the routing process of MPC. The following lemma shows that each routing round of MPC can be simulated by one layer of EMA-transformer.

**Lemma B.3** (Routing). *For any R-round MPC protocol $\pi$ having $q$ machines each with local memory $s$ and any $r \in [R-1]$, there exists an EMA-transformer $\texttt{route}_r$ with $H = O(s)$ heads and $m = O(\log q)$ for Q and K, $m = O(s^5 \log q)$ for V that takes input $X = \texttt{MachineIn}_r$ and produces output $\texttt{route}_r(X) = \texttt{MachineIn}_{r+1}$.*

*Proof.* Follow the assumption in [55], we encode the local computation into the element-wise operations $Q(\cdot), K(\cdot), V(\cdot)$ of transformer. The main part of the proof will focus on using EMA to route $\texttt{MachineOut}_r$ to $\texttt{MachineIn}_{r+1}$.

We assign a unique positional encoding or identifier to each machine $i$, denoted by $p_i$. This can be done with $O(\log q)$ bits. This encoding serves as a unique key to retrieve the message in each machine. The high level idea is to create a query for each machine $i$ and a key for each $\texttt{dest} \in \texttt{sent}_i$ and the associated value is the message $\texttt{Msg}_{\texttt{dest}}^i$ sent to $\texttt{dest}$ in the protocol. Since each machine can send at most $s$ messages to other machines, we create $s$ EMA heads and each head is responsible for one message for all the $q$ machines. Each machine retrieves the message sent to them by having a query in each head. Because each query can attend only to perfectly matching keys, each distinct outbound message must be passed by a different attention head, but multiple inbound messages may be received by the same attention head.

Specifically, let $Q^h, K^h, V^h$ be the query, key, value embedding after the machine local computation for each head $h \in [s]$. Set $q_i^h = p_i$ for all $h$, so

$$Q^1 = Q^2 = \cdots = Q^s = \begin{pmatrix} p_1^\top \\ p_2^\top \\ \vdots \\ p_q^\top \end{pmatrix}.$$

Let $k_i^h = p_{\texttt{dest}_h^i}$ for $\texttt{dest}_h^i \in \texttt{sent}^i = \{\texttt{dest}_1^i, \texttt{dest}_2^i, \ldots, \texttt{dest}_s^i\}$, where $\texttt{dest}_j^i$ is the destination machine index for the $j$th word message that machine $i$ sends. The key matrices are constructed as follows:

$$K^1 = \begin{pmatrix} p_{\texttt{dest}_1^1}^\top \\ p_{\texttt{dest}_1^2}^\top \\ \vdots \\ p_{\texttt{dest}_1^q}^\top \end{pmatrix}, \quad K^2 = \begin{pmatrix} p_{\texttt{dest}_2^1}^\top \\ p_{\texttt{dest}_2^2}^\top \\ \cdots \\ p_{\texttt{dest}_2^q}^\top \end{pmatrix}, \quad \ldots, \quad K^s = \begin{pmatrix} p_{\texttt{dest}_s^1}^\top \\ p_{\texttt{dest}_s^2}^\top \\ \vdots \\ p_{\texttt{dest}_s^q}^\top \end{pmatrix}.$$

Let $v_i^h$ be some embedding of $(\mathtt{Msg}_{\mathtt{dest}_h^i}^i, \mathtt{dest}_h^i, \mathtt{i})$, denoted by $v_i^h = \mathtt{emb}_i^h(\mathtt{Msg}_{\mathtt{dest}_h^i}^i, \mathtt{dest}_h^i, \mathtt{i})$ for some $\mathtt{emb}_i^h$ defined later, and

$$
V^1 = \begin{pmatrix} \mathtt{emb}_1^1(\mathtt{Msg}_{\mathtt{dest}_1^1}^1, \mathtt{dest}_1^1, 1) \\ \mathtt{emb}_2^1(\mathtt{Msg}_{\mathtt{dest}_1^2}^2, \mathtt{dest}_1^2, 2) \\ \vdots \\ \mathtt{emb}_q^1(\mathtt{Msg}_{\mathtt{dest}_1^q}^q, \mathtt{dest}_1^q, \mathtt{q}) \end{pmatrix}, \quad \ldots, \quad V^s = \begin{pmatrix} \mathtt{emb}_1^s(\mathtt{Msg}_{\mathtt{dest}_s^1}^1, \mathtt{dest}_s^1, 1) \\ \mathtt{emb}_2^s(\mathtt{Msg}_{\mathtt{dest}_s^2}^2, \mathtt{dest}_s^2, 2) \\ \vdots \\ \mathtt{emb}_q^s(\mathtt{Msg}_{\mathtt{dest}_s^q}^q, \mathtt{dest}_s^q, \mathtt{q}) \end{pmatrix}.
$$

By such construction of $Q, K, V$, in our EMA, each query will retrieve the average value of the messages whose key exactly matches the query. However, by setting the value matrix this way, we might corrupt the message when there are more than one $k_i^h \in K^h$ that are equal to the same query. To solve this problem, we can apply the same *multiple hashing-based encoding* in Lemma 3.2 from [55], which encodes each message in multiple fixed locations generated by a sparse binary matrix and have an extra "validity bit" indicating whether the message is corrupted or not. We restate an adapted version of their Lemma 3.2 here.

**Lemma B.4** (Lemma 3.2 of [55]; message encoding in sparse averaging). *For any message size $\Delta \in \mathbb{N}$, message count bound $\alpha \in \mathbb{N}$, there exist an encoding function $\phi$ such that $\phi$ takes in $(\mathtt{Msg}_{\mathtt{dest}_h^i}^i, \mathtt{dest}_h^i, \mathtt{i})$ defined above where the size of it is bounded by $\Delta$ for all $i \in [q]$ and $h \in [\alpha]$ and encodes it into $\mathtt{emb}_i^h(\mathtt{Msg}_{\mathtt{dest}_h^i}^i, \mathtt{dest}_h^i, \mathtt{i}) \in \mathbb{R}^m$ with $m = O(\alpha^4 \Delta \log q)$, and a decoder function $\varphi$ such that $\varphi$ takes in the output of the EMA with $Q, K, V$ defined above and decodes it into $(\mathtt{Msg}_{\mathtt{dest}_h^i}^i, \mathtt{dest}_h^i, \mathtt{i})$.*

Let $\mathtt{rcvd}^i = \{\mathtt{src}_1^i, \mathtt{src}_2^i, \ldots, \mathtt{src}_s^i\}$, where $\mathtt{src}_j^i$ is the $j$th source machine index that machine $i$ receives message from. Because $|\mathtt{sent}^i| \leq s$ and $|\mathtt{rcvd}^i| \leq s$, in each head of EMA, there are at most $s$ values get retrieved and averaged for each query. Thus, here we can just apply Lemma B.4 with $\alpha = \Delta = s$, which gives us an embedding dimension bound $m = O(s^5 \log q)$. $\square$

We then show with one layer EMA-transformer, we can properly initialize the setup of MPC by converting $\mathtt{Input} = (\mathtt{input}_1, \mathtt{input}_2, \ldots, \mathtt{input}_n)$ to $\mathtt{MachineIn}_1$, the input before the first round of MPC computation, ie the input is distributed evenly on the first $\lceil \frac{n}{s} \rceil$ machines.

**Lemma B.5** (Initialization). *For any $R$-round MPC protocol $\pi$ having $q$ machines each with local memory $s$ and $n$-word input, there exists an EMA-transformer $\mathtt{init}$ with $H = 1$ head, $m = O(\log q)$ for $Q, K$ and $m = O(s)$ for $V$ that takes $\mathtt{input}$ and outputs $\mathtt{init}(\mathtt{input}) = \mathtt{MachineIn}_r$.*

*Proof.* The input should be distributed accross each machine $1 \leq i \leq \lceil \frac{n}{s} \rceil$ with $\mathtt{MachineIn}_1^i = \{(\mathtt{input}_{\mathtt{idx}}, \mathtt{idx}) : \mathtt{idx} \in \mathtt{s}(\mathtt{i}-1)+1, \ldots, \min\{\mathtt{n}, \mathtt{si}\}\}$. Let $q_{in} = \lceil \frac{n}{s} \rceil$ be the number of machines to store the initial input. Since the input given to $\mathtt{init}$ are $n$ tokens (here we treat each token as a machine), we need to rearange the memory so that the input is distributed on the first $q_{in}$ tokens.

Same as before, we use the positional encoding $p_i$ to be the unique identifier for each machine. We create a key value pair for each input token and the key corresponds to the identifier of the machine that $\mathtt{input}_{\mathtt{idx}}$ goes to and the value be $(\mathtt{input}_{\mathtt{idx}}, \mathtt{idx})$. Also, create a query for each machine $i \in [q_{in}]$.

For each machine $i \in [q_{in}]$, define the query embedding $q_i = p_i$,

$$
Q = \begin{pmatrix} p_1^\top \\ p_2^\top \\ \vdots \\ p_{q_{in}}^\top \end{pmatrix}
$$

For each token $\mathtt{input}_{\mathtt{idx}}, \mathtt{idx} \in [\mathtt{n}]$, let $\mathtt{dest}_{\mathtt{idx}} = \lceil \frac{\mathtt{idx}}{\mathtt{s}} \rceil$ be the machine storing the token, define the key embedding $k_{\mathtt{idx}} = p_{\mathtt{dest}_{\mathtt{idx}}}$,

$$
K = \begin{pmatrix} p_{\mathtt{dest}_1}^\top \\ p_{\mathtt{dest}_2}^\top \\ \vdots \\ p_{\mathtt{dest}_n}^\top \end{pmatrix}
$$

Let $i' = \mathtt{idx} \mod \mathtt{s}$. For each token $\mathtt{input_{idx}}$, $\mathtt{idx} \in [\mathtt{n}]$, define the value embedding $v_{\mathtt{idx}} \in \mathbb{R}^{2s}$ to be $(\mathtt{input_{idx}}, \mathtt{idx})$ in the $2i'-1, 2i'$-th entry and 0 in all other entry,

$$
V = \begin{pmatrix}
\mathtt{input_1} & 1 & 0 & 0 & 0 & \ldots & 0 & 0 \\
0 & 0 & \mathtt{input_2} & 2 & 0 & \ldots & 0 & 0 \\
0 & 0 & 0 & 0 & 0 & \ldots & \mathtt{input_s} & s \\
\mathtt{input_{s+1}} & s+1 & 0 & 0 & 0 & \ldots & 0 & 0 \\
\vdots & \vdots & \vdots & \vdots & \vdots & \vdots & \vdots & \vdots
\end{pmatrix}
$$

By setting the value matrix like this, we can avoid corrupting the messages. $\square$

Last, we show that with an additional one layer EMA-transformer, we can map the final round $\mathtt{MachineIn_R}$ to the $\mathtt{output}$ of MPC protocol where the output is store in the first $\lceil \frac{n}{s} \rceil$ machines.

**Lemma B.6** (Final Output). *For any R-round MPC protocol $\pi$ having $q$ machines each with local memory $s$ and $n$-word input, there exists an EMA-transformer $\mathtt{out}$ with $H = 1$ head, $m = O(\log q)$ for $Q, K$ and $m = O(s)$ for $V$ that takes $\mathtt{MachineIn_R}$ and outputs $\mathtt{out}(\mathtt{MachineIn_R})_{:n} = \pi(\mathtt{input}) = \mathtt{output}$.*

*Proof.* First, the element-wise operations can compute $\mathtt{MachineOut_R}$ from $\mathtt{MachineIn_R}$. The output is distributed accross each machine $1 \leq i \leq \lceil \frac{n}{s} \rceil = q_{out}$ with memory of machine $i$ be $\mathtt{output^i} = \{(\mathtt{output_{idx}}, \mathtt{idx}) : \mathtt{idx} \in \mathtt{s(i-1)+1}, \ldots, \min\{\mathtt{n}, \mathtt{si}\}\}$. Then, we just need to retrieve the output tokens from all the $q_{out}$ machines and distribute them back to $n$ tokens. This step does the inverse job of init. We create a query for each token $\mathtt{output_{idx}}$ for all $\mathtt{idx} \in [\mathtt{n}]$. Let $\mathtt{src_{idx}} = \lceil \frac{\mathtt{idx}}{\mathtt{s}} \rceil$ be the machine token $\mathtt{output_{idx}}$ is in.

For each token $\mathtt{output_{idx}}$, $\mathtt{idx} \in [\mathtt{n}]$, define the query embedding $q_{\mathtt{idx}} = p_{\mathtt{src_{idx}}}$,

$$
Q = \begin{pmatrix}
p_{\mathtt{src_1}}^{\mathsf{T}} \\
p_{\mathtt{src_2}}^{\mathsf{T}} \\
\vdots \\
p_{\mathtt{src_n}}^{\mathsf{T}}
\end{pmatrix}
$$

For each machine $i \in [q_{out}]$, create a key $k_i = p_i$,

$$
K = \begin{pmatrix}
p_1^{\mathsf{T}} \\
p_2^{\mathsf{T}} \\
\vdots \\
p_{q_{out}}^{\mathsf{T}}
\end{pmatrix}
$$

The value associates with each key $i$ is the memory $\mathtt{MsgOut_i}$ stored in each machine $i$. Define the value embedding $v_i = \mathtt{MsgOut_i}$,

$$
V = \begin{pmatrix}
\mathtt{MsgOut_1} \\
\mathtt{MsgOut_2} \\
\vdots \\
\mathtt{MsgOut_{q_{out}}}
\end{pmatrix}
$$

By choosing a proper element-wise function $\varphi$, $\mathtt{out}(\mathtt{MachineIn_R})_{i,1} = \mathtt{output_i}$. $\square$

The theorem follows from stacking the elements from these three lemmas. Each lemma gives us a single layer of the final EMA-transformer $T$ with embedding dimension $m = O(N^{5\varepsilon} \log N)$:

$$
T = \mathtt{out} \circ \mathtt{route_{R-1}} \circ \cdots \circ \mathtt{route_1} \circ \mathtt{init} \qquad \square
$$

**Remark B.7** (General $(\gamma, \varepsilon)$-MPC). *The above simulation works the same for $(\gamma, \varepsilon)$-MPC by padding $\max(0, O(N^{1+\gamma-\varepsilon}) - N)$ empty chain-of-thought tokens in the input.*

**Remark B.8** (Number of heads). *The standard transformer can simulate MPC using the same embedding dimension but only $1$ attention head [53, 55]. Here the EMA needs $O(N^\varepsilon)$ heads, and we leave how to improve the number of heads for future work.*

Since EM attention is a special case of ANN attention with $r = 0$ and $c \to \infty$, the simulation of MPC with ANNA-transformer naturally follows from Theorem B.2.

**Corollary B.9** (ANNA simulates MPC). *For constant $0 < \varepsilon < 1$, any deterministic $R$-round MPC protocol $\pi$ with $N$ machines with $s = O(N^\varepsilon)$ words local memory, there exists an ANNA-transformer $T$ with depth $L = R + 1$, number of heads $H = O(N^\varepsilon)$, and embedding dimension $m = O(N^{5\varepsilon} \log N)$, such that $T(\texttt{input}) = \pi(\texttt{input})$ for all $\texttt{input} \in \mathbb{Z}_{2^p}^N$.*

Theorem B.2 only gives us an MPC simulation in the sublinear local memory regime when $s = O(N^{1/5-\delta})$ for any $\delta > 0$. However, a lot of MPC protocol algorithms require $s = \Omega(N^{1/2})$ local, such as MPC algorithm for 3-SUM [26] and algorithms for graphs [53]. The above simulation using EMA-transformer does not yield a sublinear embedding dimension. [53] further gives a simulation of MPC with sublinear local memory using transformer with sublinear embedding dimension by simulating one round of MPC protocol with $O(1)$ layers of transformer, instead of just one layer. Their improvement also applies here.

**Theorem B.10** (EMA simulates MPC with improved embedding dimension). *For constant $0 < \varepsilon < \varepsilon' < 1$, any deterministic $R$-round MPC protocol $\pi$ with $N$ machines with $s = O(N^\varepsilon)$ words local memory, there exists an EMA-transformer of depth $L = O(R)$, number of heads $H = O(N^{\frac{\varepsilon'-\varepsilon}{4}})$ and embedding dimension $m = O(N^{\varepsilon'})$ such that $T(\texttt{input}) = \pi(\texttt{input})$ for all $\texttt{input} \in \mathbb{Z}_{2^p}^N$.*

*Proof.* The proof relies on simulating any MPC protocol by a restricted version of MPC protocol [53] which limits the number of machines each machine can send message to. Then, use a modified version of Theorem B.2 to simulate this restricted version of MPC.

**Definition B.11** (Definition 3 of [53]). *For constants $\gamma, \varepsilon, \rho > 0$, a $(\gamma, \varepsilon, \rho)$-MPC protocol is a $(\gamma, \varepsilon)$-MPC protocol with an additional constraint: in each round, each machine can only send/receive messages from $k = O(n^\rho)$ machines, while the total size of messages it can send and receive is still $s = O(N^\varepsilon)$. We refer to $k$ as the* communication capacity.

[53] gives a construction that simulates a R-round $(\gamma, \varepsilon)$-MPC protocol with $O(R)$-round $(\gamma, \varepsilon, \rho)$-MPC protocol. We restate their proposition here.

**Lemma B.12** (Proposition 24 of [53]; $(\gamma, \varepsilon, \rho)$-MPC simulates $(\gamma, \varepsilon)$-MPC). *For constants $\gamma, \varepsilon > 0$ and $\rho \in (0, \varepsilon/2)$, if function $f$ can be computed by an $R$-round $(\gamma, \varepsilon)$-MPC, it can also be computed by a $O(\frac{R(1+\gamma)^2}{\rho^2})$-round $(\gamma, \varepsilon, \rho)$-MPC protocol.*

Therefore, we just need to simulate $(\gamma, \varepsilon, \rho)$-MPC protocol using our EMA-transformer. The simulation follows the same recipe as Thereom B.2, where we have the initialization, message passing and final output phase. Since the initialization and output of $(\gamma, \varepsilon, \rho)$-MPC follow the same rule as $(\gamma, \varepsilon)$-MPC, we only need to modify the message passing part of the simulation which corresponds to the routing Lemma B.3.

**Lemma B.13** (EMA simulates $(\gamma, \varepsilon, \rho)$-MPC). *For constant $0 < \rho < \varepsilon < 1$, any deterministic $R$-round MPC protocol $\pi$ with $N$ machines with $s = O(N^\varepsilon)$ words local memory and communication capacity $k = O(N^\rho)$, there exists an EMA-transformer of depth $L = R + 1$, number of heads $H = O(N^\rho)$ and embedding dimension $m = O(N^{\varepsilon+4\rho} \log N)$ such that $T(\texttt{input}) = \pi(\texttt{input})$ for all $\texttt{input} \in \mathbb{Z}_{2^p}^N$.*

*Proof.* For the initialization and final output part, we just use the same `init` and `out` constructed in Lemma B.5 and Lemma B.6. In the routing part (Lemma B.3), because $|\texttt{sent}^i| \le \texttt{k}, |\texttt{rcvd}^i| \le \texttt{k}$, we only need $k$ heads and in each head of EMA, there are at most $k$ keys matching each query and thus at most $k$ values get averaged for a single query. Therefore, we can apply Lemma B.4 with $\alpha = k$ and $\Delta = s$, leading to an embedding dimension $m = O(N^{\varepsilon+4\rho} \log N)$. This gives us a new $\texttt{route}'_\texttt{r}$ with reduced number of heads and embedding dimension for each round $r$.

Likewise, we stack the 3 building blocks of one-layer EMA-transformer and have an $(R + 1)$-layer EMA transformer

$$T = \texttt{out} \circ \texttt{route}'_{\texttt{R}-1} \circ \cdots \circ \texttt{route}'_1 \circ \texttt{init}$$

and this finishes the construction for the lemma. $\square$

Let $\rho = \min(\varepsilon/2, (\varepsilon' - \varepsilon)/4)$. In this setting, $\gamma = \varepsilon$. By Lemma B.12, we can simulate the $R$-round $(\gamma, \varepsilon)$-MPC by an $R' = O(\frac{R(1+\varepsilon)^2}{\min(\varepsilon^2, (\varepsilon'-\varepsilon)^2)})$-round $(\gamma, \varepsilon, \rho)$-MPC. Then, by Lemma B.13, we can simulate this $R'$-round $(\gamma, \varepsilon, \rho)$-MPC by an $R' + 1$-layer EMA transformer with $O(N^\rho)$ heads and embedding dimension $O(N^{\varepsilon+4\rho} \log N) = O(N^{\varepsilon'})$. $\qquad\square$

Again, the improved simulation result of ANNA-transformer follows from Theorem B.10.

**Corollary B.14** (ANNA simulates MPC with improved embedding dimension). *For constant* $0 < \varepsilon < \varepsilon' < 1$*, any deterministic $R$-round MPC protocol $\pi$ with $N$ machines with $s = O(N^\varepsilon)$ words local memory, there exists an ANNA-transformer of depth $L = O(R)$, number of heads $H = O(N^{\frac{\varepsilon'-\varepsilon}{4}})$ and embedding dimension $m = O(N^{\varepsilon'})$ such that $T(\texttt{input}) = \pi(\texttt{input})$ for all $\texttt{input} \in \mathbb{Z}_{2^p}^N$.*

## C  MPC can Simulate ANNA-transformer

As a warm-up, we first simulate EMA-transformers using MPC, and then generalize it to the simulation of ANNA-transformers.

**Theorem C.1** (MPC simulates EMA). *Fix constants $0 < \varepsilon < \varepsilon' < 1$. For any L-layer EMA-transformer $T$ with $mH = O(N^\varepsilon)$, there exists a $O(\frac{L}{\varepsilon'-\varepsilon})$-round MPC protocol $\pi$ with local memory $s = O(N^{\varepsilon'})$ and $P = O(N^{1+\varepsilon-\varepsilon'})$ machines such that $\pi(\texttt{input}) = T(\texttt{input})$ for all $\texttt{input} \in \mathbb{Z}_{2^p}^N$.*

*Proof.* We first show how to use MPC to simulate one layer of EMA-transformer. In the high level, for each token $x_i$, we have a token Machine $i$ which is responsible for computing the key, query and value embedding for $x_i$ and other element-wise computation on $x_i$. The main bulk of the proof is to search for the exact matching keys for each query and send the averaged values associated with the matching keys to the token machines. In order to do this, we sort all the key and value pairs $(k_i, v_i)$ in the order defined by the key. We divide the sorted key and value pairs into buckets such that each bucket contains the same keys. For each bucket, we have a "meta-info" machine to store the indices of the machines that contains the keys in the bucket. We then compute the averaged values within each bucket and store the averaged value into the "meta-info" machine and propagate the value to all the queries that match with the key.

To begin with, let $\mathcal{X}$ denote the space of query and key, and we define a comparator $<$ over $\mathcal{X}$ in order to sort. Without loss of generality, we just define it to be the lexicographical ordering comparator. Based on this comparator, we define a query ranking permutation of $[N]$ by $\sigma = (\sigma_1, \sigma_2, \ldots, \sigma_N)$ and a key ranking permutation of $[N]$ by $\sigma' = (\sigma'_1, \sigma'_2, \ldots, \sigma'_N)$ such that

$$q_{\sigma_1} < q_{\sigma_2} < \cdots < q_{\sigma_N}$$
$$\text{and } k_{\sigma'_1} < k_{\sigma'_2} < \cdots < k_{\sigma'_N}$$

For the "meta-info" machine, we use a uniform hash function $h : \mathcal{X} \to [N]$ to map queries and keys to their corresponding "meta-info" machine. Recall that for a uniform hash function $h$, $\mathbb{P}(h(a) = h(b)) = \frac{1}{N}$, for any $a, b \in \mathcal{X}$ and $a \neq b$. Therefore,

$$\mathbb{P}(\exists i \text{ such that the size of bucket } h(q_i) \geq s)$$
$$\leq \mathbb{P}(\exists s \text{ different elements fall into one bucket})$$
$$\leq \binom{N}{s} \frac{1}{N^s} \leq \frac{1}{s!} = \frac{1}{N^{\varepsilon'}!}$$

With high probability, each "meta-info" machine is responsible for at most $s$ keys or queries.

We divide the machines into different types and summarize the role of each type of machine here:

- For $i \in [N]$, Machine $i$ is the *token machine* for $x_i$. This machine performs all the element-wise computation for token $i$. Specifically, it computes the query, key, value embeddings $q_i, k_i, v_i$ and element-wise operations after the attention layer.

- For $i \in [\lceil mN/s \rceil]$, Machine $(i, Q)$ is a data structure machine for sorting queries and storing the $i$th chunk of the sorted list of queries after sorting. In other words, let $n_q = \lfloor s/m \rfloor$ be the number of queries each machine can store and, at the end of sorting, machine $(i, Q)$ stores $\{q_{\sigma_{(i-1) \cdot n_q + 1}}, \ldots, q_{\sigma_{i \cdot n_q}}\}$.

- For $i \in \lceil 2mN/s \rceil$, Machine $(i, KV)$ is a data structure machine for sorted list of key and value pairs. In other words, let $n_k = \lfloor s/2m \rfloor$ be the number of key and value pairs each machine can store and, at the end of sorting, machine $(i, KV)$ stores $\{(k_{\sigma'_{(i-1) \cdot n_q + 1}}, v_{\sigma'_{(i-1) \cdot n_q + 1}}), \ldots, (k_{\sigma'_{i \cdot n_q}}, v_{\sigma'_{i \cdot n_q}})\}$.

- For $i \in [N]$, Machine $(i, h_q)$ is the "meta-info" machine for the queries whose hash value is $i$. Let $h_q^i = \{q_j | j \in [N], h(q_j) = i\}$. This machine stores the location information of $q \in h_q^i$ in the sorted list. Specifically, for all $q \in h_q^i$, this machine stores the start machine index, i.e. (start, Q) where start $= \arg\min_j \{q \in \text{Machine } (j, Q)\}$, and the end machine index, i.e. (end, Q) where end $= \arg\max_j \{q \in \text{Machine } (j, Q)\}$.

- For $i \in [N]$, Machine $(i, h_k)$ is the "meta-info" machine for the keys whose hash value is $i$. Let $h_k^i = \{k_j | j \in [N], h(k_j) = i\}$. This machine stores the location information of $k \in h_k^i$ in the sorted list. Specifically, for all $k \in h_q^i$, this machine stores the start machine index, i.e. (start, KV) where start $= \arg\min_j \{k \in \text{Machine } (j, KV)\}$, and the end machine index, i.e. (end, Q) where end $= \arg\max_j \{k \in \text{Machine } (j, KV)\}$.

- The auxiliary machines needed for message propagation.

We proceed to discuss the MPC protocol for computing the output of one layer single head EM-attention transformer. In the first round, (same as the token dispersion stage of [55]), route each token $x_i$ to its corresponding token machine $i$.

In the second round, each token machine $i$ computes the query, key value embedding $q_i = Q(x_i), k_i = K(x_i), v_i = V(x_i)$ and sends $(q_i, i)$ to the sorting query data structure machine $(\lceil mi/s \rceil, Q)$ and $(k_i, v_i, i)$ to the sorting key data structure machine $(\lceil 2mi/s \rceil, KV)$.

Then, sorting query data structure machines $((i, Q)$ for all $i \in \lceil mN/s \rceil)$ sorts the queries. Sorting in MPC has been well studied, and this can be done in constant number of rounds [23].

**Lemma C.2** (MPC Sorting). *There exists an MPC protocol with local memory $s = O(N^{\varepsilon'})$ that can sort $N$ items and each item has size $O(N^\varepsilon), \varepsilon < \varepsilon'$ in $O(\frac{1}{\varepsilon' - \varepsilon})$ rounds with $O(N^{1 + \varepsilon - \varepsilon'})$ machines.*

After sorting, for each $i \in [N]$, we need to send the location information of $q_i$, ie which data structure machines contains $q_i$, to its "meta-info" machine $(h(q_i), h_q)$. The idea is to build an $\frac{s}{m}$-ary tree structure to aggregate the information and each query data structure machine is a leaf node of this tree. Recall that each machine $(i, Q)$ stores the queries $S = \{q_{\sigma_{(i-1) \cdot n_q + 1}}, \ldots, q_{\sigma_{i \cdot n_q}}\}$. If $S$ contains the start and end of a particular query vector $q_l$, then $(i, Q)$ sends a message $(q_l, (i, Q))$ to machine $(h(q_l), h_q)$. Machine $(i, Q)$ also sends the first and last query to its parent machine in the tree, i.e. sends the messages $(q_{\sigma_{(i-1) \cdot n_q + 1}}, (i, Q), \text{first})$ and $(q_{\sigma_{i \cdot n_q}}, (i, Q), \text{last})$. After the parent node collects all the messages from the leaf node, it then does the same as its child: if it contains the start and end of a certain query $q$, it sends to the location information (the first and last machine that store it) of the query to its corresponding "meta-info" machine $(h(q), h_q)$, and it sends the first and last query and their location information to its parent machine. This is done recursively, and since there are $\lceil mN/s \rceil$ query data structure machines in total, the depth of this $\frac{s}{m}$-ary tree is $O(\log_{s/m} mN/s) = O(\frac{1}{\varepsilon' - \varepsilon})$, which means $O(\frac{1}{\varepsilon' - \varepsilon})$ rounds and $O(mN/s)$ machines suffice.

We do the same for $(k, v)$ pairs. The sorting data structure machines $(i, KV)$ sort the (k, v) pairs based on the order of $k$. As before, we build a $\frac{s}{2m}$-ary tree to send the location information to the "meta-info" machine of each key. The different part from query is that we combine the values that have the same key. For each machine in the $\frac{s}{2m}$-ary tree, it computes the averaged value associated with each key it contains and sends the averaged value to the corresponding "meta-info" machine. In particular, for each $k_i$, the 'meta-info" machine for $k_i$, $(h(k_i), h_k)$, contains the information $(k_i, \bar{v})$ where $\bar{v}$ is the average of $v_j$'s such that $k_j = k_i$.

Next, the "meta-info" machines of query and key need to exchange information to retrieve the corresponding value for each query. Each $(i, h_k)$ sends the $(k, \bar{v})$ pairs it has to the machine $(i, h_q)$. Then, each $(i, h_q)$ machine matches the $q$ and $k$, and sends the associated value $\bar{v}$ to the $q$. Note that this step can be done by back propagating the $\frac{s}{m}$-ary tree constructed for sending the location information of $q$ to $(h(q), h_q)$. In other words, we can just reverse the message sending direction in this tree. Therefore, each query in the query data structure machine receives the value it retrieves and from each query data structure machine $(i, Q)$, we can send the retrieved value for each query to its corresponding token machine, which is the inverse of the second round.

To summarize, the total rounds needed is $O(\frac{1}{\varepsilon' - \varepsilon})$ and the number of machines needed is $O(mN/s) = O(N^{1+\varepsilon-\varepsilon'})$. To make this work for $H$ heads, we can create $H$ copies of this and each copy runs in parallel. Since $mH = O(N^\varepsilon)$, the bounds for number of rounds and machines still hold. By creating this MPC simulation for each of the $L$-layers, we stack them in the order of layers yielding the complete simulation for $L$-layer EMA-transformer. $\qquad\square$

Next, we generalize the above algorithm and proceed to simulate the ANN attention that can be computed by Algorithm 1. Since Algorithm 1 is a randomized algorithm, we assume that the MPC protocol shares all the random seeds needed for all the layers of ANNA-transformer.

**Theorem C.3** (MPC simulates ANNA). *Fix constants $0 < \varepsilon < \varepsilon' < 1$. For any $L$-layer ANNA-transformer $T$ (as implemented by Algorithm 1) with $mH = O(N^\varepsilon)$, there exists a $O(L/(\varepsilon' - \varepsilon))$-round MPC protocol $\pi$ with local memory $s = O(N^{\varepsilon'})$ and $P = O(N^{1+\varepsilon-\varepsilon'+3/c^2})$ machines such that $\pi(\texttt{input}) = T(\texttt{input})$ for all $\texttt{input} \in \mathbb{Z}_{2^p}^N$.*

*Proof.* The high level idea of simulating ANNA-transformer is very similar to simulating EMA. We have the same kinds of machines as before. The biggest difference is that, instead of having one hash table for queries and keys, we now have $\ell$ hash tables, one for each round of hashing, and we sort the queries and keys based on the hash values of queries and keys. Again, we first outline different types of machines we will use.

- For $i \in [N]$, machine $i$ is the *token machine* for $x_i$. This machine performs all the element-wise computation for token $i$. Specifically, it computes the query, key, value embeddings $q_i, k_i, v_i$ and element-wise operations after the attention layer.

- For $i \in [\lceil mN/s \rceil], t \in [\ell]$, machine $(i, Q, h^t)$ is a data structure machine for sorted queries for the $t$-th hash table and the $i$-th chunk of the sorted list of queries, where the ordering of sorting is based on $g_t(q)$ from Algorithm 1.

- For $i \in \lceil 2mN/s \rceil$, Machine $(i, KV, h^t)$ is a data structure machine for sorted list of key and value pairs for the $t$-th hash table and the $i$-th chunk of the sorted list of key and value pairs, where the ordering of sorting is based on $g_t(k)$.

- For $i \in [N], t \in [\ell]$, Machine $(g_t(q_i), h_q, t)$ is the "meta-info" machine for the queries whose $t$-th hash value is $g_t(q_i)$. Let $h^t_{q_i} = \{q_j | j \in [N], g_t(q_j) = g_t(q_i)\}$. This machine stores the location information of $q \in h^t_{q_i}$ in the $t$-th hash table. Specifically, for all $q \in h^t_{q_i}$, this machine stores the start machine index, i.e. $(\text{start}, Q, h^t)$ where $\text{start} = \arg\min_j \{q \in \text{Machine } (j, Q, h^t)\}$, and the end machine index, i.e. $(\text{end}, Q, h^t)$ where $\text{end} = \arg\max_j \{q \in \text{Machine } (j, Q, h^t)\}$.

- For $i \in [N], t \in [\ell]$, Machine $(g_t(k_i), h_k, t)$ is the "meta-info" machine for the keys whose $t$-th hash value is $g_t(k_i)$. Let $h^t_{k_i} = \{k_j | j \in [N], g_t(k_j) = g_t(k_i)\}$. This machine stores the location information of $k \in h^t_{k_i}$ in the $t$-th hash table. Specifically, for all $k \in h^t_{k_i}$, this machine stores the start machine index, i.e. $(\text{start}, KV, h^t)$ where $\text{start} = \arg\min_j \{k \in \text{Machine } (j, KV, h^t)\}$, and the end machine index, i.e. $(\text{end}, Q, h^t)$ where $\text{end} = \arg\max_j \{k \in \text{Machine } (j, KV, h^t)\}$.

- The auxiliary machines needed for message propagation.

Like before, we still use each token machine to compute the embeddings $q_i, k_i, v_i \in \mathbb{R}^m$. Then, each token machine need to send $(q_i, i)$ and $(k_i, v_i, i)$ to the data structure machines, machine $(\lceil mi/s \rceil, Q, h^t)$ and machine $(\lceil 2mi/s \rceil, KV, h^t)$, for all $t \in \ell$. Because $\ell = N^{3\rho}$, we use the $\frac{s}{m}$-ary

tree to propagate the queries and keys to the corresponding data structure machines. This takes $O(\frac{1}{\varepsilon'-\varepsilon})$ rounds and $O(N^{1+3\rho+\varepsilon-\varepsilon'})$ machines.

Then, for each query hash table $t \in [\ell]$, the data structure machines sort the queries based on the hash value of the queries. Same as Theorem C.1, we use the $\frac{s}{m}$-ary tree to send the location information of each hash bucket to its corresponding "meta-info" machine. For each key, value pair hash table $t \in [\ell]$, the data structure machines sort the key, value pairs based on the hash value of the keys. After that, use the $\frac{s}{2m}$-ary tree to propagate the information to the corresponding "meta-info" machine. The difference from the EMA simulation is that each machine in this $\frac{s}{2m}$-ary tree maintains the sum of values whose key has the same hash values instead of the averaged value, and also maintains a count of the number of keys. These can be done in $O(\frac{1}{\varepsilon'-\varepsilon})$ rounds and $O(N^{1+3\rho+\varepsilon-\varepsilon'})$ number of machines.

Next, the key "meta-info" machine send the sum of values and count to the corresponding query "meta-info" machines, i.e. machine $(g_t(k_i), h_k, t)$ sends to machine $(g_t(q_i), h_q, t)$. Each query "meta-info" machine then follows the $\frac{s}{m}$-ary tree, broadcasting the sum of values and counts to the queries in the hash table. finally, each query in the hash table needs to propagate the information back to its original token machine. Since each token machine will receive message from $\ell = N^{3\rho}$ machines, we again reverse the $\frac{s}{m}$-ary tree that send the query to each data structure machine. During the aggregation, each machine in the $\frac{s}{m}$-ary tree still maintains the sum of values and the sum of counts it receives. After receiving the sum of values and counts, each token machine $i$ then calculates $\text{ANNA}(q_i) = $ sum of the values divided by the counts.

The above simulates one layer of ANNA-transformer in $O(\frac{1}{\varepsilon'-\varepsilon})$ rounds and using $O(N^{1+3\rho+\varepsilon-\varepsilon'})$ machines, where $\rho = 1/c^2$. Therefore, by stacking the simulation for $L$ layers, this gives $O(\frac{L}{\varepsilon'-\varepsilon})$ rounds in total. To extend to $H$ heads, we just need to instantiate the above simulation for $H$ parallel copies and because $mH = O(\varepsilon)$, the total number of rounds and machines still remains the same. □

# D   ANN/EM Attention can simulate low-rank Attention via MPC

We simulate the low-rank attention using ANN attention by first giving a MPC algorithm for computing low-rank attention and then convert it to ANNA-transformer.

**Theorem D.1** (ANNA/EMA simulates low-rank Attention). *For constants $0 < \varepsilon < \varepsilon' < 1$, any low-rank attention based transformer with depth $L$, rank $r$, embedding dimension $m$ and $rm = O(N^\varepsilon)$ can be simulated by an EMA/ANNA-transformer with depth $O(\frac{L}{\varepsilon'-\varepsilon})$, number of heads $H = O(N^{\varepsilon'})$ and embedding dimension $m = O(N^{5\varepsilon'} \log N)$.*

*Proof.* We prove this theorem by first proving that any one-layer of low-rank attention can simulated by constant number of rounds of MPC.

**Lemma D.2** (MPC simulates low-rank Attention). *For constants $0 < \varepsilon < \varepsilon' < 1$, any one-layer low-rank attention with rank $r$, embedding dimension $m$ and $rm = O(N^\varepsilon)$ can be simulated by a $O(\frac{1}{\varepsilon'-\varepsilon})$-round MPC protocol with local memory $s = O(N^{\varepsilon'})$ and $O(N)$ machines.*

*Proof.* Assume $rm = O(N^\varepsilon)$ and local memory of MPC $s = O(N^{\varepsilon'})$ where $\varepsilon < \varepsilon'$. Same as what we do in MPC simulating EMA, for each token $x_i, i \in [N]$, we have a token machine $i$ to compute the embedding of $x_i$ but we need to compute it in the kernel space, i.e. $q_i = Q'(x_i), k_i = K'(x_i)$ and $v_i = V(x_i)$. To compute $K'(X)^\intercal V(X)$, recall that

$$K'(X)^\intercal V(X) = \sum_{i=1}^{N} k_i v_i^\intercal$$

We just need to compute the sum of $N$ matrices of size $r \times m$. Each token machine $i$ computes the matrix $k_i v_i^\intercal$ and we construct a $\lfloor \frac{s}{rm} \rfloor$-ary tree of machines to compute the sum. The leaves of the tree are all the token machines and each node is responsible for computing the sum of $\lfloor \frac{s}{rm} \rfloor$ number of matrices. We know from the previous simulation that the depth of the tree is $O(\frac{1}{\varepsilon'-\varepsilon})$. After we obtain the matrix $M = K'(X)^\intercal V(X) \in \mathbb{R}^{r \times m}$, in order to compute $Q(X)K'(X)^\intercal V(X)$, we just

need to propagate the matrix $M$ to all the token machines. And each token machine $i$ computes $q_i^\mathsf{T} M$. By reversing the direction of message propagation in the computing sum tree, we can propagate $M$ to all the token machines in $O(\frac{1}{\varepsilon'-\varepsilon})$ rounds. Therefore, we can simulate kernel attention with $O(\frac{1}{\varepsilon'-\varepsilon})$ rounds in total. $\qquad\square$

For $L$ layers of low-rank attention transformer, we construct the MPC for each layer using Lemma D.2 and again we use the local computation of each token machine to simulate the element-wise computation. We stack the $L$ MPCs together, which has $O(\frac{L}{\varepsilon'-\varepsilon})$ rounds. The theorem follows from applying Theorem B.2 and Corollary B.9. $\qquad\square$

Since the core of the proof is through MPC simulating low-rank Attention, we can also apply Theorem B.10 and Corollary B.14 which simulate MPC with better embedding dimension to get a improved embedding dimension for simulating low-rank attention transformer.

**Corollary D.3** (ANNA/EMA simulates low-rank Attention with improved embedding dimension)**.** *For constants $0 < \varepsilon < \varepsilon' < 1$, any low-rank attention based transformer with depth $L$, rank $r$, embedding dimension $m$ and $rm = O(N^\varepsilon)$ can be simulated by an EMA/ANNA-transformer with depth $O(\frac{L}{(\varepsilon'-\varepsilon)\cdot\min(\varepsilon^2,(\varepsilon'-\varepsilon)^2)})$, number of heads $H = O(N^{\frac{\varepsilon'-\varepsilon}{4}})$ and embedding dimension $m = O(N^{\varepsilon'})$.*

# E   Discussion on Reformer

We formally define Reformer as a computational model here.

**Definition E.1** (Reformer attention)**.** *Given query, key, value embeddings $Q(X), K(X), V(X) \in \mathbb{R}^{N \times m}$ such that $q_i := k_i = Q(X)[i,:] = K(X)[i,:], v_i = V(X)[i,:]$,* Reformer *attention proceeds as follows:*

1. *Apply a hash function $h : \mathbb{R}^m \to U$ on $\{q_1, \ldots, q_N\}$;*

2. *Sort all $q_i$'s (and thus $k_i$'s) by $h(q_i)$ and partition all $q_i$'s into chunks of size $B \leq O(1)$, and let $h'(q_i)$ be the label of the chunk that $q_i$ is in (the queries in each chunk can have different hash values);*

3. *For each $q_i$, only attend to $k_j$'s such that they are in the same chunk.*

*The output embedding for $q_i$ is therefore*

$$\sum_{j:h'(k_j)=h'(q_i)} \frac{\exp(\langle q_i, k_j \rangle)}{\displaystyle\sum_{j':h'(k_{j'})=h'(q_i)} \exp(\langle q_i, k_{j'} \rangle)} \cdot v_j.$$

We define $f_\ell : [N] \to [N]^B$ as the function that specifies the set of keys each query should compute inner product with in the $\ell$-th layer. From the Reformer constraints, we have $\forall i \in [N]$:

1. $f_\ell(i) = \{a_1, a_2, \ldots, a_B\} \in [N]^B$ is a set (no repetition).
2. $i \in f_\ell(i)$.
3. For any $j \in f_\ell(i)$, $f_\ell(j) = f_\ell(i)$.

In the $\ell$-th layer attention computation for each query $q_i$, Reformer computes

$$\sum_{j \in f_\ell(i)} \frac{\exp(\langle q_i, k_j \rangle)}{\displaystyle\sum_{j' \in f_\ell(i)} \exp(\langle q_i, k_{j'} \rangle)} \cdot v_j.$$

We first study a restricted version of Reformer that fix the communication pattern beforehand i.e. $f_\ell$ is input-independent for all $\ell \in [L]$, and show that it can not compute the sum of all the input tokens.

**Definition E.2** (SUM). *Given input $X = (x_1, x_2, \ldots, x_N), x_i \in [M]$, and $M = N^{O(1)}$, the SUM task is defined as $\mathrm{SUM}(X) = \sum_{i=1}^{N} x_i$. We say a Reformer $T$ computes $\mathrm{SUM}$ if for all $X$, $T(X)_N = \mathrm{SUM}(X)$.*

Here $T(X)_N$ is the $N$-th output of T given the input $X$. One can choose any position to be the final output position, and here WLOG we choose the last token to follow the autoregressive generation model convention.

**Proposition E.3.** *Fix $L = O(1)$ and $\{f_\ell\}_{\ell=1}^{L}$. Any Reformer $T$ with $L$ layers and each layer the attention pattern is specified by $\{f_\ell\}_{\ell=1}^{L}$ can not compute $\mathrm{SUM}(X)$: there exists an $X$, $|T(X)_N - \mathrm{SUM}(X)| \geq \epsilon$, for any $0 < \epsilon < M/2$.*

*Proof.* We denote each layer's element-wise computation by $\{\phi_\ell\}_{\ell=1}^{L}$. Let $T^\ell(X)_i$ denote the $i$-th output of $T$ after $\ell$ layers of computation. We prove this proposition by induction.

Inductive hypothesis: $T^\ell(X)_i$ is a function of at most $B^\ell$ different $x_i \in X$.

Base case: $\ell = 1$

$$T^1(X)_i = \sum_{j \in f_1(i)} \frac{\exp(\langle q_i, k_j \rangle)}{\displaystyle\sum_{j' \in f_1(i)} \exp(\langle q_i, k_{j'} \rangle)} \cdot v_j$$

$$= \sum_{j \in f_1(i)} \frac{\exp(\langle Q(x_i), K(x_j) \rangle)}{\displaystyle\sum_{j' \in f_1(i)} \exp(\langle Q(x_i), K(x_{j'}) \rangle)} \cdot V(x_j)$$

$$= \phi_1(x_{a_1}, x_{a_2}, \ldots, x_{a_B}) \text{ where } f_1(i) = \{a_1, \ldots, a_B\}$$

which is a function of at most $B$ $x_i$'s in $X$.

Inductive step: consider

$$T^{\ell+1}(X)_i = \sum_{j \in f_{\ell+1}(i)} \frac{\exp(\langle Q(T^\ell(X)_i), K(T^\ell(X)_j) \rangle)}{\displaystyle\sum_{j' \in f_{\ell+1}(i)} \exp(\langle Q(T^\ell(X)_i), K(T^\ell(X)_{j'}) \rangle)} \cdot V(T^\ell(X)_j)$$

$$= \phi_{\ell+1}(T^\ell(X)_{a_1}, \ldots, T^\ell(X)_{a_B}) \text{ where } f_1(i) = \{a_1, \ldots, a_B\}$$

Since each $T^\ell(X)_{a_j}$ is a function of at most $B^\ell$ variables from $X$, $T^{\ell+1}(X)_i$ is a function of at most $B \cdot B^\ell = B^{\ell+1}$ variables from $X$.

Therefore, if $T^L(X)_i$ is a function of all $\{x_1, \ldots, x_N\}$, we need $B^L \geq N$ and thus $L = \Omega(\log_B N)$. In the case $B = O(1)$ and $L = O(1)$, $T^L(X)_i$ is a function of $B^L \ll N$ variables. WLOG, consider $T^L(X)_N$ is a function of $\{x_1, \ldots, x_{B^L}\}$. Then, $x_{B^L+1}$ can be any number in $[M]$ that makes $T^L(X)_N$ far from $\mathrm{SUM}(X)$. $\square$

Therefore, if Reformer has any power, it must come from the sorting part, because the sorting algorithm have access to the information of all the token inputs.

Although constant-layer Reformer can not compute SUM, one can easily show that one layer of ANNA-transformer can compute SUM by setting $v_i = N x_i$ and $k_1 = k_2 = \cdots = k_N = q_N$ for all $i \in [N]$, thereby retrieving all the $v_i$'s and averaging them.

# F  ANNA-transformer Solves $k$-hop and Match2

## F.1  ANNA/EMA-transformer Solves Match2

**Theorem F.1.** *For any $N, M = N^{O(1)}$, there exists an EMA-transformer $T$ with one layer, one attention head, and embedding dimension 1 such that $T(X) = \mathrm{Match2}(X)$ for all $X \in [M]^N$.*

*Proof.* Given input $X \in [0, M]^{N \times 1}$. Let $Q(X) = \phi(X)Q, K(X) = \phi(X)K, V(X) = \phi(X)V$, where $Q, K, V$ are matrices in $\mathbb{R}^{2 \times 1}$. Define $\phi$ by $\phi(x) = (x, 1)$ and

$$Q = \begin{pmatrix} 1 \\ 0 \end{pmatrix}, K = \begin{pmatrix} -1 \\ M \end{pmatrix}, V = \begin{pmatrix} 0 \\ 1 \end{pmatrix}$$

such that

$$\phi(X)Q = \begin{pmatrix} x_1 \\ x_2 \\ \vdots \\ x_N \end{pmatrix}, \phi(X)K = \begin{pmatrix} M - x_1 \\ M - x_2 \\ \vdots \\ M - x_N \end{pmatrix}, \phi(X)V = \begin{pmatrix} 1 \\ 1 \\ \vdots \\ 1 \end{pmatrix}.$$

As a result, for each $1 \le i \le N$, if there exists $1 \le j \le N$ such that $x_i + x_j = M$, then

$$((\phi(X)Q)(\phi(X)K)^{\mathsf{T}})[i, j] = \frac{1}{|\{j \in [N] : x_i + x_j = M\}|}.$$

Otherwise, $((\phi(X)Q)(\phi(X)K)^{\mathsf{T}})[i, j] = 0$ for all $1 \le j \le N$. Finally, we can calculate that if $|\{j \in [N] : x_i + x_j = M\}| \ne 0$, then

$$\text{EMA}(\phi(X)Q, \phi(X)K, \phi(X)V)[i] = \frac{1}{|\{j \in [N] : x_i + x_j = M\}|} \cdot |\{j \in [N] : x_i + x_j = M\}| = 1,$$

and if $|\{j \in [N] : x_i + x_j = M\}| = 0$, then

$$\text{EMA}(\phi(X)Q, \phi(X)K, \phi(X)V)[i] = 0. \qquad \square$$

That gives the same result for ANNA-transformers.

**Corollary F.2.** *For any $N, M = N^{O(1)}$, there exists an ANNA-transformer $T$ with one layer, one attention head, and embedding dimension 1 such that $T(X) = \text{Match2}(X)$ for all $X \in [M]^N$.*

### F.2 ANN transformer Solves $k$-hop

We first show that ANNA transformers can solve induction head (1-hop).

**Lemma F.3.** *Fix constants $0 < \varepsilon < \varepsilon' < 1$, and $|\Sigma| \le N$. There exists an ANNA-transformer $T$ with $L = O(\frac{1}{\varepsilon \cdot (\varepsilon' - \varepsilon)^2})$ layers, $H = O(N^{(\varepsilon' - \varepsilon)/4})$ heads per layer, and embedding dimension $m = O(N^{\varepsilon'})$ such that $T(w)_i = w_{\sigma(w,i)}$, if $\sigma(w, i) \ne 0$; $T(w)_i = \bot$, if $\sigma(w, i) = 0 \; \forall i \in [N]$, for all $w \in \Sigma^N$.*

*Proof.* We prove this lemma by designing a constant-round MPC algorithm with local memory $s = O(N^\varepsilon)$ and $N/s$ machines to solve 1-hop. Since $|\Sigma| \le N$, each token can be embedded with $O(\log N)$ bits ($O(1)$ words). Denote the input $w^N = (x_1, x_2, \ldots, x_N)$. The MPC algorithm works as following:

1. For each $x_i$, retrieve the next token $x_{i+1}$ and each token on the machine is stored as the embedding of $(x_i, i, x_{i+1}, i + 1)$.

2. Define a comparator $<$ for the object $(x_i, i, x_{i+1}, i + 1)$. For two tuples $(x_i, i, x_{i+1}, i + 1)$ and $(x_j, j, x_{j+1}, j + 1)$, if $x_i \ne x_j$, then $x_i < x_j \Rightarrow (x_i, i, x_{i+1}, i + 1) < (x_j, j, x_{j+1}, j + 1)$; if $x_i = x_j$, then $i < j \Rightarrow (x_i, i, x_{i+1}, i + 1) < (x_j, j, x_{j+1}, j + 1)$. Sort $(x_i, i, x_{i+1}, i + 1)$ by the comparator $<$.

3. Each token $(x_i, i, x_{i+1}, i + 1)$ in the sorted list retrieves the token before it in the sorted list, denoted by $(x_j, j, x_{j+1}, j + 1)$. Update the embedding of token: if $x_j = x_i$, the embedding of the token $x_i$ becomes $(i, x_{j+1}, j + 1)$ i.e. $(i, w_{\sigma(w,i)}, \sigma(w, i))$; if $w_j \ne w_i$, then the embedding of the token $x_i$ becomes $(i, \bot, 0)$.

4. Send each $(i, w_{\sigma(w,i)}, \sigma(w, i))$ to the correct output machine $\lceil \frac{i}{s} \rceil$ and output $w_{\sigma(w,i)}$ for token $i$.

For step 1, each machine only needs to send message to its neighbor machine: machine $i$ sends message to machine $i - 1$, and this only takes 1 round. In step 2, each tuple is only $O(\log N)$ bits, so by Lemma C.2, the sorting takes $O(\frac{1}{\varepsilon})$ rounds. In step 3, again each machine only needs to send message to its neighbor machine: machine $i$ sends message to machine $i + 1$, and this only takes 1 round. In step 4, each machine for the sorted list sends at most $s$ tuples stored in it to the correct output machine which takes 1 round. Thus, the MPC algorithm has $O(\frac{1}{\varepsilon})$ rounds in total.

Then, we convert this MPC algorithm to an ANNA-transformer. By Corollary B.14, this gives us an ANNA-transformer with number of heads $H = O(N^{(\varepsilon' - \varepsilon)/4})$, embedding dimension $m = O(N^{\varepsilon'})$ and number of layers $L = O(\frac{1}{\varepsilon \cdot (\varepsilon' - \varepsilon)^2})$. $\qquad\square$

Now we show that ANNA-transformers can solve $k$-hop with $O(\log k)$ layers.

**Theorem F.4.** *Fix constants $0 < \varepsilon < \varepsilon' < 1$, $|\Sigma| \le N$ and any $k \in \mathbb{N}$. There exists an ANNA-transformer $T$ with $L = O\left(\frac{1}{\varepsilon \cdot (\varepsilon' - \varepsilon)^2} + \frac{\log k}{(\varepsilon' - \varepsilon)^2}\right)$ layers, $H = O(N^{(\varepsilon' - \varepsilon)/4})$ heads per layer, and embedding dimension $m = O(N^{\varepsilon'})$ such that $T(w)_i = w_{\sigma^k(w, i)}, \forall i \in [N]$, for all $w \in \Sigma^N$.*

*Proof.* We prove this theorem by constructing an $O(\log k)$-rounds MPC with $s = O(N^\varepsilon)$ local memory and $O(N/s)$ machines. We show that this MPC algorithm can compute $k$-hop by induction. Let $k = \sum_{j=0}^{\lfloor \log k \rfloor} k_j 2^j$ and $k_{:\ell} = \sum_{j=0}^{\ell-1} k_j 2^j$, where $k_j \in \{0, 1\}$.

Inductive hypothesis: after $O(\frac{1}{\varepsilon}) + 2\ell$ rounds of MPC computation, the token embedding for each token $i$ encodes the information of this tuple

$$(i, w_{\sigma^{2^\ell}(w, i)}, \sigma^{2^\ell}(w, i), w_{\sigma^{k:\ell}(w, i)}, \sigma^{k:\ell}(w, i))$$

Base case: $\ell = 0, k = 1$ is implied by Lemma F.3. After step 3, we have $(i, w_{\sigma(w, i)}, \sigma(w, i))$. Now consider $k = \ell + 1$. For each $i$, the machine (Machine $\lceil i/s \rceil$) that contains $(i, w_{\sigma^{2^\ell}(w, i)}, \sigma^{2^\ell}(w, i), w_{\sigma^{k:\ell}(w, i)}, \sigma^{k:\ell}(w, i))$ sends the message $(i, \sigma^{2^\ell}(w, i))$ to machine that contains $\sigma^{2^\ell}(w, i)$ as the first entry of the tuple which is machine $\lceil \frac{\sigma^{2^\ell}(w, i)}{s} \rceil$. Machine $\lceil \frac{\sigma^{2^\ell}(w, i)}{s} \rceil$ then send the following tuple to machine $\lceil i/s \rceil$:

$$(\sigma^{2^\ell}(w, i), w_{\sigma^{2^\ell}(w, \sigma^{2^\ell}(w, i))}, \sigma^{2^\ell}(w, \sigma^{2^\ell}(w, i)), w_{\sigma^{k:\ell}(w, \sigma^{2^\ell}(w, i))}, \sigma^{k:\ell}(w, \sigma^{2^\ell}(w, i)))$$
$$= (\sigma^{2^\ell}(w, i), w_{\sigma^{2^{\ell+1}}(w, i)}, \sigma^{2^{\ell+1}}(w, i), w_{\sigma^{k:\ell+2^\ell}(w, i)}, \sigma^{k:\ell+2^\ell}(w, i))$$
$$:= (\sigma^{2^\ell}(w, i), t_1, t_2, t_3, t_4)$$

Since each machine has at most $s$ tuples and the function $\sigma(w, i)$ is one-to-one except for $\perp$, the number of messages each machine sends and receive is bounded by $s$. After machine $\lceil i/s \rceil$ receiving the above message, it update the tuple for the token $i$:

1. if $k_\ell = 0$, token $i$ is updated as: $(i, t_1, t_2, w_{\sigma^{k:\ell}(w, i)}, \sigma^{k:\ell}(w, i))$

2. if $k_\ell = 1$, token $i$ is updated as: $(i, t_1, t_2, t_3, t_4)$

By definition, the embedding of token $i$ now is:

$$(i, w_{\sigma^{2^{\ell+1}}(w, i)}, \sigma^{2^{\ell+1}}(w, i), w_{\sigma^{k:\ell+1}(w, i)}, \sigma^{k:\ell+1}(w, i))$$

The above inductive step only takes 2 rounds of MPC. Therefore, the total round is $O(\frac{1}{\varepsilon}) + 2(\ell + 1)$. When $\ell = \lfloor \log k \rfloor + 1$, this algorithm compute the output for $k$-hop.

Again, we can convert this MPC algorithm to an ANNA-transformer. By Corollary B.14, this gives us an ANNA-transformer with number of heads $H = O(N^{(\varepsilon' - \varepsilon)/4})$, embedding dimension $m = O(N^{\varepsilon'})$ and number of layers $L = O(\frac{1}{\varepsilon \cdot (\varepsilon' - \varepsilon)^2} + \frac{\log k}{(\varepsilon' - \varepsilon)^2})$. $\qquad\square$

# G Experimental details

Here are the details of the experimental setup. All the experiments are launched on 2 GPUs: NIVIDIA Titan RTX and NVIDIA Titan Xp.

We train a modified version of the attention matrix and then distill from the trained model using ANNA implemented by the angular distance LSH family from [6]. Our softmax attention normalizes all the queries and keys in $Q(X)$ and $K(X)$ to have unit norm, and computes $\text{softmax}(\beta \cdot Q(X)K(X)^\top)V(X)$ with a hyperparameter $\beta > 0$.

## G.1 Match2 experiments

**Dataset generation.** Inspired by the way [37] generating data for Match3 task, a triple-wise version of Match2, we generate the data for Match2 using the same algorithm but change to pair-wise relation when computing the label. Each sample is a tuple $(X, Y)$, where $X = (x_1, x_2, \ldots, x_N)$, and each $x_i$ is an integer sampled from $\{1, 2, \ldots, 36\}$; $Y = (y_1, y_2, \ldots, y_N)$, and each $y_i = \mathbb{1}\{\exists j \centerdot x_i + x_j = 0 \bmod 37\}$. The sequence length $N$ is set to 32. When sampling the data, we ensure that each batch is balanced by having the distribution of one's in $Y$ the same: each batch has 4 bins and each bin corresponds to each percentage $[0, 25\%), [25\%, 50\%), [50\%, 75\%), [75\%, 100\%]$ of one's in $Y$; each bin size is $1/4$ of the batch size. See Algorithm 3 for details.

---

**Algorithm 3** Match2 Dataset Generation

---

**Input:** $N = 32$, $M = 37$, dataset size $D$
**Output:** Dataset of $(X, Y)$ pairs
1: Initialize 4 empty bins for ones-percentage ranges: $[0, 25), [25, 50), [50, 75), [75, 100]$
2: $N_b \leftarrow D/4$
3: **while** total examples in bins $< D/40$ **do**
4:     Sample $X \in \{1, \ldots, M\}^N$ uniformly at random
5:     Calculate the percentage $p$ of one's in $Y$
6:     **if** size of the bin $p$ is in $< N_b$ **then**
7:         Add $(X, Y)$ to the correct bin
8: **for** each bin **do**
9:     **while** size of bin $< N_b$ **do**
10:         Randomly sample $(X, Y)$ from bin
11:         Sample permutation $\pi$ over $[0, \ldots, N - 1]$
12:         $X^* \leftarrow X[\pi]$, $Y^* \leftarrow Y[\pi]$
13:         Add $(X^*, Y^*)$ to bin
14: Combine and shuffle all bins into final dataset
15: **return** Dataset

---

**Training details.** We trained 3 models with $\beta \in \{0.1, 1, 10\}$ respectively, with Adam optimizer on cross-entropy loss and learning rate 0.01. Each model has one layer, one attention head, embedding dimension $m = 64$ and an MLP with width $4m$ and GeLU activation. The dataset size, batch size, training steps are $10000, 32, 20000$ respectively.

We apply ANNA with number of hash tables $\ell \in \{1, 2, \ldots, 16\}$ and number of hash functions for each table $z \in \{1, 2, \ldots, 6\}$ on all the 3 trained models, and $\beta = 0.1$ has the best performance (error can be 0). Since the implementation of ANNA is randomized, for each combination of $(\ell, z)$, we run 10 times and report the averaged error over the 10 runs. See Figure 1a for plotted performance when $\beta = 0.1$. In this setting, $\ell \geq 8, z = 1$ can achieve 0 error on the test set with 256 test samples.

## G.2 Induction heads experiments

**Dataset generation.** We use the data generation algorithm from [55]. Each sample $(X, Y)$ is of the form $X = (k, X')$ for $k \in \{0, 1\}$ for training samples and $k = 1$ for test samples, $X' \in \Sigma^{N-1}$, $Y = (0, Y')$, where $Y'$ is the $k$-hop label for $X'$. Here the sequence length $N = 100$ and alphabet size $|\Sigma| = 4$.

**Training details.**    We trained 3 models with $\beta \in \{0.1, 1, 10\}$ respectively, with Adam optimizer on cross-entropy loss and learning rate $0.01$. Each model has $2$ layers, each layer with one attention head, embedding dimension $m = 128$ and an MLP with width $4m$ and GeLU activation. We use online training: at each training step, sample fresh new data to train. The batch size and training steps are $32, 400000$ respectively.

We apply ANNA on all the 3 trained models. For the first layer, the number of hash tables $\ell$ is chosen from $\{32, 40, 48, \dots, 96\}$ and $z$ is chosen from $\{1, 2, 3, 4\}$. For the second layer, the number of hash tables $\ell$ is chosen from $\{4, 8, 12, \dots, 32\}$ and $z$ is chosen from $\{1, 2, 3, 4\}$. When evaluating the test error, we compute the error on all the tokens. Note that this is different from [55], where they only compute the error on the tokens whose induction head exists to avoid overestimating the performance when $k$ is large and has a large fraction of null outputs. In our setting, $k = 1$ which doesn't have many null outputs, and it is important for the model to learn when to output the null token.

We found $\beta = 1$ has the best performance, so we report it in Figure 1b. Again, the errors are averaged over $10$ runs for each combinations and taken the minimum over $z$'s. One can see that $32$ hash tables in the first layer and $4$ hash tables in the second layer already gives highly non-trivial performance: the error rate is $0.2$ over $100$ samples and each sample has $100$ token predictions, while random guess would give $0.75$ error rate. With more hash tables in the first layer, the error rate can go below $0.1$.

