# OpenReview forum: "Fast attention mechanisms: a tale of parallelism"
_NeurIPS.cc/2025/Conference — NeurIPS 2025 poster_

### Official Review · Reviewer_sfjv · 2025-07-05

**Clarity:** 3
**Significance:** 3
**Originality:** 3
**Rating:** 5
**Confidence:** 2

**Summary:**

This paper studies Approximate Nearest Neighbor Attention (ANNA), a subquadratic transformer mechanism that matches the expressiveness of vanilla attention. More precisely, they show an equivalence between ANNA and MPC algorithms (Theorem 1.1) and an equivalence between ANNA and low-rank transformers (Theorem 1.2).

The motivation for their work is that prior work showed a coarse correspondence between transformers and MPC algorithms. However, this correspondence required $N^2$ machines to simulate 1 layer of attention.  A key selling point of the paper is that ANNA narrows this gap to an almost linear dependency on the number of machines.

In Section 2, the authors define the main objects: standard attention, transformers, and recall what MPC algorithms are. In Section 3, the authors introduce ANNA attention mechanism. In Section 4, the authors state their main results: they first show a correspondence between ANNA-transformers and MPC (Theorems 4.1 and 4.2)  and then use this link to compare ANNA with other sub-quadratic attention schemes like low-rank transformers (Theorem 4.4). Lastly, in Section 5, the authors show that ANNA-transformers can solve reasoning tasks such as Match-2, induction heads and k-hops.

**Questions:**

I listed my questions and remarks in the strengths and weaknesses section.

**Ethical Concerns:**

["NO or VERY MINOR ethics concerns only"]

**Limitations:**

My main concern about this paper is the absence of a limitations section.

**Paper Formatting Concerns:**

No concern.

**Quality:**

3

**Strengths And Weaknesses:**

I am not an expert at all in the field related to this paper. Therefore, I apologize in advance for my low-confidence assessment.

**Strengths**:

Even though I am not a specialist in this area, the core contribution is clear and significant: ANNA attention tightens the transformer–MPC correspondence from Θ(N²) to almost-linear (in N) number of MPC machines.
The motivation for ANNA is natural: Definition 3.3 abstracts the essential locality property (“attend only to r-near keys”) without tying it to one concrete algorithm.
Unification of the two main “fast-attention” families: Theorem 4.4 shows that every low-rank transformer can be simulated by an ANNA transformer. Researchers can therefore analyse a single model (nearest-neighbour attention) instead of a zoo of unrelated approximations.




**Weaknesses**:

Below I list the main questions or concerns that, in my view, the authors should still address. But again, I have very little understanding in the field.

- **Novelty of the paper?**: the ingredients ­– LSH-based attention and the correspondence transformers MPC – are known. The paper’s contribution is to tighten this correspondence from $O(N^2)$ to $O(N^{1+o(1)}$ machines and, through this, to prove that ANNA strictly subsumes low-rank attention. It would help if the authors highlighted more explicitly which lemmas are new, e.g. the almost-linear-machine simulation (Theorem 4.2) and the reduction from low-rank to ANNA (Theorem 4.4).
- **The authors should add a limitation section**: the experiments distill from a soft-max model; the paper does not yet demonstrate an end-to-end differentiable training scheme for ANNA, although such schemes exist for related LSH attentions (Reformer, Routing Transformer).
- **Memory & constant factors**: in line 230, the authors say that the runtime is sub-quadratic in $O( N^{1+3\rho}) $. With c=10 (which is already pretty aggressive), we get $\rho=0.01$, so the exponent is 1.03 and $\ell = N^{0.03} log N$. For N=16 000 that is 64 hash tables per layer – still a noticeable overhead, though far from the “900 tables” suggested by a literal reading of the bound. A discussion of how these constants affect wall-clock speed and memory would be valuable.

---

> ### Author Rebuttal · Authors · 2025-07-30
>
> Thank you for your review. We appreciate all the feedback provided.
>
> > Novelty of the paper?: the ingredients – LSH-based attention and the correspondence transformers MPC – are known. The paper’s contribution is to tighten this correspondence from $O(N^2)$ to $O(N^{1+o(1)})$ machines and, through this, to prove that ANNA strictly subsumes low-rank attention. It would help if the authors highlighted more explicitly which lemmas are new, e.g. the almost-linear-machine simulation (Theorem 4.2) and the reduction from low-rank to ANNA (Theorem 4.4).
>
> Thanks for your suggestion! We will highlight these connections more in the main results part of the paper. The ANNA simulation of MPC (Theorem 4.1) is new, because ANNA is a more restrictive and different model from transformer, and thus the correspondence between transformer and MPC does not apply to ANNA. Put differently, we propose ANNA not as an approximator to standard transformers (implemented via, say, LSH), but rather as an alternative, more efficient model in itself (and hence the model to be trained to start with).
>
> > The authors should add a limitation section: the experiments distill from a soft-max model; the paper does not yet demonstrate an end-to-end differentiable training scheme for ANNA, although such schemes exist for related LSH attentions (Reformer, Routing Transformer).
>
> Thank you for the suggestion; we will add a limitation section discussing these issues. Note that Reformer and Routing Transformer are also not differentiable, so they also do not have end-to-end differentiable training schemes. (Cf. "Differentiable Approximations for Distance Queries" by Abdelkader and Mount, which gives a scheme for approximate differentiable distance query computations that is efficient in low-dimensional spaces.) But it is a beautiful open question!
>
> > Memory & constant factors: in line 230, the authors say that the runtime is sub-quadratic in $O(N^{1+3\rho})$. With c=10 (which is already pretty aggressive), we get $\rho=0.01$, so the exponent is 1.03 and $\ell=N^{0.03}\log N$. For N=16 000 that is 64 hash tables per layer – still a noticeable overhead, though far from the “900 tables” suggested by a literal reading of the bound. A discussion of how these constants affect wall-clock speed and memory would be valuable.
>
> Our theoretical analysis is not fine-grained enough to pin down the constants suppressed by the big-O notation, so wall-clock time and the precise number of hash tables is beyond the scope of our paper. Please also see our reply to Reviewer XSKf regarding (lack of) memory overhead.
>
> We'd love to discuss more if you have any further questions.

---

> > ### Comment · Reviewer_sfjv · 2025-08-04
> >
> > I thank the authors for their response, that I found pretty convincing. Unfortunately, I have limited background on this topic but the arguments of the authors make sense. I maintain my score.

---

### Official Review · Reviewer_1KAS · 2025-07-05

**Clarity:** 3
**Significance:** 3
**Originality:** 3
**Rating:** 5
**Confidence:** 3

**Summary:**

This paper studies the ability of the transformer architecture to simulate the MPC model when the standard attention mechanism in the transformer is replaced with sparse attention based on Approximate Nearest Neighbor. The authors show that ANNA transformers (i.e. transformers that use Approximate Nearest Neighbor Attention) have the same expressive power as standard transformers in terms of simulating the MPC. In addition, the authors show that ANNA transformers may be strictly more powerful than transformers that use low-rank attention, in the sense that there are problems that are easy for ANNA but hard for low-rank attention, and ANNA transformers can simulate low-rank attention based transformer. Finally, the authors show that ANNA transformers can solve some canonical reasoning tasks such as Match2 and k-hop with near-optimal depth. Overall, this paper extends prior study on the transformer's representational capacity (with respect to simulating MPC algorithms) to an interesting case where the attention matrix is sparse.

**Questions:**

Line 355-356, I think there is an error. In this example, sigma(w, 9) = 7, so sigma(w, sigma(w,9)) = sigma(w, 7) = 6, and hence w_{sigma(w, sigma(w,9))} = a, not b.

**Ethical Concerns:**

["NO or VERY MINOR ethics concerns only"]

**Final Justification:**

After reading the author response as well as the feedback from other reviewers, I would like to keep my score and recommend acceptance. I do agree (with other reviewers) that additional practical experiments would be extremely valuable and make this work stronger.

**Limitations:**

yes

**Quality:**

3

**Strengths And Weaknesses:**

The paper is very well-written and easy to follow. The theoretical results are a nice addition to our understanding of transformer's representational capacity in terms of simulating MPC algorithms, when the attention matrix is restricted to be sparse and computable in sub-quadratic time.

The accompanying empirical results in the appendix are short, but nonetheless they validate some of the theoretical results in the paper.

On the weak side, as in prior work [51,53], this paper assumes that the element-wise operations (Q, K, V, ψ) can be arbitrary functions. This deviates from the typical transformer architectures that are used in practice.

---

> ### Author Rebuttal · Authors · 2025-07-30
>
> Thank you for your review, and we appreciate the suggested change.
>
> > On the weak side, as in prior work [51,53], this paper assumes that the element-wise operations (Q, K, V, ψ) can be arbitrary functions. This deviates from the typical transformer architectures that are used in practice.
>
> Such assumptions are necessary to establish an equivalence to MPC, since the MPC model also allows arbitrary computation by each machine on their local memory content. This is a reasonable abstraction because our goal is to understand the capabilities/limitations due to the attention (or attention-like) mechanism. (That said, many concrete MPC algorithms do have a simple local algorithm.)
>
> > Line 355-356, I think there is an error. In this example, sigma(w, 9) = 7, so sigma(w, sigma(w,9)) = sigma(w, 7) = 6, and hence w_{sigma(w, sigma(w,9))} = a, not b.
>
> Thank you for catching this error! We will fix the example in the final version of the paper.
>
> We'd love to clarify more you have any further questions.

---

> > ### Comment · Reviewer_1KAS · 2025-08-05
> >
> > I'd like to thank the authors for their response. I agree with other reviewers that this is a good contribution. I will keep my score and recommend accepting this work.

---

### Official Review · Reviewer_XSKf · 2025-07-07

**Clarity:** 3
**Significance:** 4
**Originality:** 3
**Rating:** 5
**Confidence:** 3

**Summary:**

In this paper, the authors introduce Approximate Nearest Neighbor
Attention (ANNA), a sub-quadratic attention mechanism for transformers
designed to improve scalability without sacrificing expressive
power. In particular, they first establish a tight connection between
ANNA-transformers and the Massively Parallel Computation (MPC)
model. More precisely, every MPC protocol with R rounds and sub-linear
local memory can be simulated by an ANNA-transformer with O(R) layers
and a sub-linear number of heads and embedding dimension; and every
ANNA-transformer with L layers can be simulated by an MPC protocol
with O(L) rounds. Then the authors prove that each low-rank
attention-transformer with L layers can be simulated by an
ANNA-transformer with O(L) layers.

To show the usefulness of ANNA-transformers, the authors use this
architecture in two concrete reasoning tasks. More precisely, the
authors provide constructions of ANNA-transformers for these tasks
that nearly match the efficiency achievable by standard transformers,
and they provide empirical evidence that ANNA-transformers can be
trained to approximately solve thes e tasks.

**Questions:**

Q1 Could you discuss the practical implications of key architectural
parameters in ANNA, such as the memory overhead of locality sensitive
hashing tables, the failure probability, and the trade-offs involved
in selecting the approximation factor c?

Q2 Do you have any empirical results on real-world benchmarks or
large-scale datasets to support the practical effectiveness of ANNA
beyond the current proof-of-concept experiments?

**Ethical Concerns:**

["NO or VERY MINOR ethics concerns only"]

**Final Justification:**

After the authors’ response, I am convinced that this paper makes a significant contribution, so I keep my acceptance score.

**Limitations:**

Yes

**Paper Formatting Concerns:**

No formatting issues.

**Quality:**

3

**Strengths And Weaknesses:**

**Strong Points**

S1 The paper establishes a tight connection between ANNA-transformers
and MPC, improving the previous connections that had larger gaps in
machine usage for simulating attention.

S2 The paper provides an algorithm to implement ANNA in sub-quadratic
time, which is based on using locality sensitive hashing.

S3 The paper provides a unified framework to reason about low-rank and
nearest neighbor approaches to efficient attention.

**Weak Points**

W1 The empirical evaluation is limited, and it mostly serves as
proof-of-concept. Performance evaluation on real-world benchmarks or
large-scale datasets is not provided.

W2 The paper does not discuss in practical terms the implications of
important factors for the architecture such as memory overheads of
locality sensitive hashing tables, failure probabilities, and
trade-offs involving the approximation factor c for ANNA.

---

> ### Author Rebuttal · Authors · 2025-07-30
>
> Thank you for the review. We appreciate all the feedback and questions provided.
>
> > W2 The paper does not discuss in practical terms the implications of important factors for the architecture such as memory overheads of locality sensitive hashing tables, failure probabilities, and trade-offs involving the approximation factor c for ANNA.
>
> > Q1 Could you discuss the practical implications of key architectural parameters in ANNA, such as the memory overhead of locality sensitive hashing tables, the failure probability, and the trade-offs involved in selecting the approximation factor c?
>
> Memory overhead: In fact, Algorithm 1 can be implemented using linear memory ($O(N)$) with the same time complexity. Instead of maintaining $\ell$ hash tables, one can just store 1 hash table of size $O(N)$ with each entry responsible for tracking the values for each query. For each round of hashing ($\ell$ rounds in total), hash all queries using the hash functions and creates empty buckets for them. Then, hash each key, and if the key hashes to an existing query bucket, its value is added (along with a count). After processing keys, each query accumulates the values and counts from its corresponding bucket. We will add a remark about the memory complexity and this linear memory algorithm to the final version of the paper.
>
> Failure probability: The failure probability decays at a polynomial rate with respect to the context length $N$. Context length $N$ is our main scaling parameter for our asymptotic analysis, so the failure probability tends to zero rapidly with $N$. (Note that $N=128000$ for GPT-4 and $N=1000000$ for Gemini.)
>
> Trade-offs for selecting approximation factor: whether there are any trade-offs for selecting $c$ is highly problem-dependent. For example, for the Match2 and k-hop tasks in this paper, even a very large $c$ (e.g., $c=1000$) works based on the theoretical results in this paper. However, for problems that are conjectured to require quadratic time, such as Orthogonal Vectors, even $c=1.1$ is probably bad.
>
> > W1 The empirical evaluation is limited, and it mostly serves as proof-of-concept. Performance evaluation on real-world benchmarks or large-scale datasets is not provided.
>
> > Q2 Do you have any empirical results on real-world benchmarks or large-scale datasets to support the practical effectiveness of ANNA beyond the current proof-of-concept experiments?
>
> The primary goal of our paper is to make theoretical contributions to the formal understanding of transformers and related architectures, and, as a result, we did not perform empirical studies on real-world benchmarks or large-scale datasets.
>
> We'd love to discuss more if you have any further questions.

---

### Decision · Program_Chairs · 2025-09-17

**Decision:**

Accept (poster)

**Comment:**

This paper introduces Approximate Nearest Neighbor Attention (ANNA), a subquadratic transformer mechanism that preserves the expressiveness of standard attention, which usually requires Quadratic time. Specifically, the authors establish an equivalence between ANNA and Massively parallel computation. The paper demonstrates the empirical validity on two reasoning tasks. Overall this paper should be of interest to  Neurips community.